# *Rai1* frees mice from the repression of active wake behaviors by light

**Shanaz Diessler[1], Corinne Kostic[2], Yvan Arsenijevic[2], Aki Kawasaki[1,2], Paul Franken[1]\***

[1]Center for Integrative Genomics, University of Lausanne, Lausanne, Switzerland; [2]Jules-Gonin Eye Hospital, Fondation Asile des Aveugles, University of Lausanne, Lausanne, Switzerland

**Abstract** Besides its role in vision, light impacts physiology and behavior through circadian and direct (*aka* 'masking') mechanisms. In Smith-Magenis syndrome (SMS), the dysregulation of both sleep-wake behavior and melatonin production strongly suggests impaired non-visual light perception. We discovered that mice haploinsufficient for the SMS causal gene, *Retinoic acid induced-1* (*Rai1*), were hypersensitive to light such that light eliminated alert and active-wake behaviors, while leaving time-spent-awake unaffected. Moreover, variables pertaining to circadian rhythm entrainment were activated more strongly by light. At the input level, the activation of rod/cone and suprachiasmatic nuclei (SCN) by light was paradoxically greatly reduced, while the downstream activation of the ventral-subparaventricular zone (vSPVZ) was increased. The vSPVZ integrates retinal and SCN input and, when activated, suppresses locomotor activity, consistent with the behavioral hypersensitivity to light we observed. Our results implicate *Rai1* as a novel and central player in processing non-visual light information, from input to behavioral output.

## Introduction

In addition to its evident importance for vision, light is essential for multiple physiological functions. For instance, light is the most potent environmental signal ensuring correct entrainment of endogenously generated circadian rhythms to the external 24 hr day. In mammals, light information encoded in the retina and received by the suprachiasmatic nuclei (SCN), which harbor the master circadian pacemaker, can reset circadian rhythm phase of overt behaviors such as sleep and locomotor activity (*Brainard and Hanifin, 2005*). Apart from these circadian effects, light also directly impacts physiology, behavior, and cognition, including the well-known light suppression of the circadian hormone melatonin, as well as the maybe less well known light-induced increases in body temperature, cortisol, heart rate, alertness, performance, and mood (*Badia et al., 1991*; *Cajochen et al., 1992*, *Cajochen et al., 2000*; *Leproult et al., 2001*; *Lewy et al., 1980*; *Scheer et al., 2004*; *Stephenson et al., 2012*). These activating effects of light are observed in diurnal species, like us, while in nocturnal species, like the mouse, light favors sleep and suppresses locomotor activity (*Aschoff and von Goetz, 1988*). In patients diagnosed with Smith-Magenis syndrome (SMS), despite unimpaired vision, a number of symptoms strongly suggest that non-visual light perception is compromised. Whether these effects are direct or mediated through the circadian system is currently unknown. The aim of the current study was therefore to take advantage of an SMS mouse model to elucidate which of the pathways encoding light information were underlying some of the behavioral and physiological characteristics of the disease.

SMS is a multiple congenital anomalies/mental retardation disorder associated with a heterozygous interstitial deletion on chromosome 17p11.2 in >95% of patients (*Edelman et al., 2007*; *Greenberg et al., 1991*; *Smith et al., 1986*), or rarely with point mutations in *Retinoic acid induced*

**\*For correspondence:** paul.franken@unil.ch

**Competing interests:** The authors declare that no competing interests exist.

1 (*Rai1*), a gene located within this deletion (*Slager et al., 2003*). Common phenotypic features include obesity, self-injurious, aggressive behavior, as well as distinct craniofacial and skeletal abnormalities (*Edelman et al., 2007*). Among the cardinal symptoms are sleep disturbances, including early sleep on- and offset, repeated and prolonged nocturnal awakening, as well as excessive daytime sleepiness (*Smith et al., 1998a*). Sleep disturbances during the night are likely to contribute not only to daytime sleepiness, but also to temper tantrums, hyperactivity, and attention deficits during the day (*Smith et al., 1986*, *Smith et al., 1998b*; *De Leersnyder et al., 2003*). In conjunction with these sleep-wake disturbances, remarkably, the time course of melatonin levels shows an inverted rhythm in SMS patients, peaking during daytime and with low concentrations at night (*De Leersnyder, 2006*; *Potocki et al., 2000*), lending further support to the notion of a circadian misalignment and/or impaired light perception.

Because patients with *Rai1* mutations display most of the SMS features (*Edelman et al., 2007*; *Potocki et al., 2003*), including the sleep/circadian phenotype, *Rai1* haplo-insufficiency is thought to be causal (*Smith et al., 1998a*; *Dubourg et al., 2014*). A mouse model for SMS has been engineered and *Rai1* haplo-insufficient mice recapitulate some of the SMS features such as obesity and craniofacial phenotypes (*Bi et al., 2005*). Although aspects of circadian rhythms of locomotor activity have been evaluated in these mice (*Lacaria et al., 2013*), a comprehensive analysis of the effects of light on sleep-wake behavior and locomotor activity is lacking. We therefore determined whether circadian rhythmicity, entrainment, or direct light effects contribute to the SMS phenotype, by recording locomotor and electro-encephalogram (EEG) activity to evaluate sleep and wakefulness under a number of lighting conditions. To gain insight into the pathways contributing to the extreme light-driven suppression of locomotor activity that we discovered, we evaluated the anatomical and functional integrity of the retina and the neuronal activation after a light pulse at the level of the SCN and the ventral subparaventricular zone (vSPVZ), which both play important roles in regulating active wake behaviors. Paradoxically, our study revealed that the acute and sustained hypersensitivity to light at the behavioral level was accompanied by a reduced response to light at the retinal and SCN levels. Only at the level of the vSPVZ was the behavioral hypersensitivity to light accompanied by an increased activating response to light.

## Results

The expression and timing of sleep are regulated by three main processes: circadian time keeping, sleep homeostasis, and the response to environmental light (*Cajochen et al., 1992*; *Dijk and Czeisler, 1995*). Here, we addressed the proper functioning of all three processes in *Rai1* haploinsufficient mice. While probing the circadian timing system, we discovered that the suppression of locomotor activity by light was importantly increased in *Rai1*$^{+/-}$ mice. Below we will present these findings first before presenting the circadian phenotypes we observed. Subsequently, by recording the electroencephalogram (EEG), we were able to extend the prominent 'locomotor-activity-suppression-by-light' phenotype to include EEG-defined active wake behavior and performed two experiments to discern whether these exacerbated direct effects of light concerned the acute or the sustained effects of light on behaviour and the EEG. EEG analyses was also used to verify whether altered sleep homeostasis; that is, differences in the compensatory response to sleep loss, contributed to the disease phenotype. We end by presenting anatomical data concerning retinal integrity and functional data on the acute, activating effects of light at the retinal, SCN, and vSPVZ levels.

### Increased suppression of locomotor activity by light

We first asked whether *Rai1* plays a role in circadian clock function and entrainment by assessing patterns of spontaneous locomotor activity both under standard light-dark conditions (LD12:12) and constant dark (DD) conditions. Under LD12:12 conditions, mice of both genotypes were more active during the dark compared to the light period, reaching highest activity levels in the first 6 hr after dark onset (ZT12-18; *Figure 1A,B*), consistent with the mouse being a nocturnal species. Despite this overall similarity of the distribution of locomotor activity over the 24 day, we observed a number of striking differences between the two genotypes. Whereas during the light period *Rai1*$^{+/+}$ mice displayed occasional activity bouts, especially immediately preceding dark onset and following light onset, activity in *Rai1*$^{+/-}$ mice was restricted to the dark with an abrupt increase in activity immediately following dark onset and an equally abrupt suppression of activity after light onset, leaving the

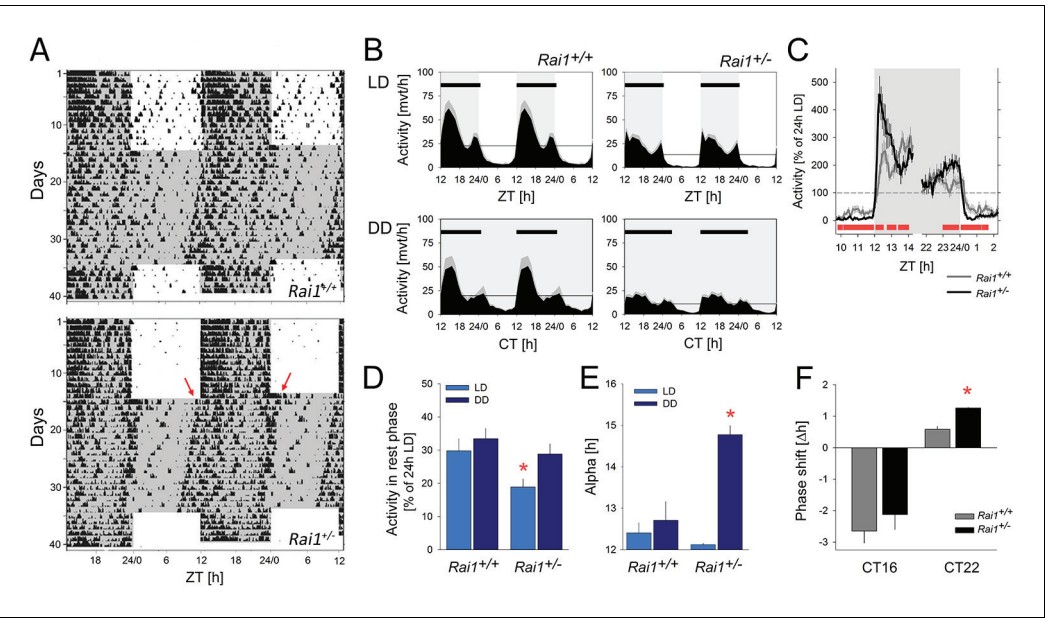

**Figure 1.** Marked suppression of locomotor activity by light in $Rai1^{+/-}$ mice. (**A**) Representative double-plotted actograms of one wild-type $Rai1^{+/+}$ (top) and one $Rai1+/-$ mouse (bottom panel), under 14 days of LD12:12 and 20 days of constant darkness (DD). Grey areas mark darkness. Note that in $Rai1^{+/-}$ mice activity is restricted to the dark periods under LD12:12 and that after the transition into DD its active period (alpha) expands with an earlier onset and a delayed offset (red arrows) of activity. (**B**) Average profiles of hourly activity values (black area with SEM in grey) under LD12:12 (top) and DD (bottom panels) for $Rai1^{+/+}$ (left; n = 11) and $Rai1^{+/-}$ (right panels; n = 12). Light-grey areas denote periods of darkness. Horizontal black bars indicate mean active periods (alpha or α; see also Panel E). Note the alpha expansion under DD and overall lower activity in $Rai1^{+/-}$ mice. Horizontal lines indicate average 24 hr activity levels for each genotype under each condition. The LD12:12 average activity levels were used in Panels C and D to correct for genotype differences in overall activity levels. (**C**) The suppression of activity by light in $Rai1^{+/-}$ mice under LD12:12 was abrupt and the relative increase in activity immediately following dark onset and immediately preceding light onset were ~2 fold larger than in wild-type mice. Values were expressed as % of each animal's 24 hr average activity under LD12:12 conditions (=100% dashed horizontal line). (**D**) Even after correcting for individual differences in overall activity (set to 100% as in C), activity in the 12 hr light periods remains lower in $Rai1^{+/-}$ mice compared to wild-type littermates (light blue bars; p<0.05, t-test) and also compared to their own activity during the rest phase under DD (rho or ρ; dark blue bars; *p<0.01, *paired t-test*). (**E**) Comparison of alpha under LD12:12 and DD conditions. Alpha expansion was observed only in $Rai1^{+/-}$ mice (*p<0.001, *paired t-test*). (**F**) A light pulse at CT22 elicited a ~ 1 hr larger phase advance in $Rai1^{+/-}$ mice compared to wild-type littermates (*p<0.005, *t-test*), while a light pulse administered at CT16 resulted in the expected phase delay. Note that this experiment was performed in a separate cohort of mice ($Rai1^{+/-}$ n = 4; $Rai1^{+/+}$ n = 5). Error bars represent SEMs.

The following source data and figure supplement are available for figure 1:

**Source data 1.** Excel file with data sheets corresponding to each figure panel containing the numerical values on which *Figure 1* and *Figure 1—figure supplement 1* are based.

**Figure supplement 1.** The effects of lowering light intensity on duration of the active phase (alpha) and locomotor activity.

light period almost devoid of any activity (*Figure 1A,C*). The number of movements during the light phase was reduced by 67% in $Rai1^{+/-}$ mice, which is substantial considering that in the wild-type littermate controls locomotor activity reaches its lowest levels during this period (*Figure 1B*).

The overall 24 hr activity levels (both under LD and DD) were lower in $Rai1^{+/-}$ mice compared to wild-type ($Rai1^{+/+}$ *vs.* $Rai1^{+/-}$: LD12:12: 545 ± 79 *vs.* 317 ± 36; p=0.019; DD: 473 ± 77 vs. 268 ± 35 movements/24 hr; p=0.006, t-tests). These lower overall activity levels could not, however, explain the extremely low activity levels observed during the light period in $Rai1^{+/-}$ mice. Even after taking

these overall genotype differences into account by expressing activity during the light as a percentage of each animal's individual average 24 hr activity level, activity in $Rai1^{+/-}$ mice remained greatly reduced compared to wild-type littermates (p=0.022, t-test; *Figure 1D*, see also *Figure 5—figure supplement 2*). Activity during the light phase was also lower when compared to the average activity levels reached in the rest phase (*rho*) under DD conditions in $Rai1^{+/-}$ mice (p=0.007, paired t-test; *Figure 1D*) but not in $Rai1^{+/+}$ mice. Moreover, relative activity levels (again expressed as % of each animal's 24 hr activity under LD12:12) during rho under DD, did no longer differ between genotypes (*Figure 1B,D*; p=0.3, *t-test*), constituting further evidence that the loss of one $Rai1$ allele leads to a marked suppression of activity during light exposure.

The suppression of activity by light depends on its intensity (*Aschoff, 1960*) and the extreme suppression by light observed in $Rai1^{+/-}$ mice might occur at higher light intensities only. We therefore exposed a subset of mice to a low light intensity (i.e., 0.6 cds/m$^2$) compared to our habitual 6.6 cds/m$^2$. Although at these lower light levels $Rai1^{+/-}$ mice were more active compared to the higher light level condition, relative locomotor activity during the 12 hr light phase (as % of each animal's 24 hr activity under LD12:12 at 6.6 cdm/m$^2$) was still greatly suppressed in $Rai1^{+/-}$ mice compared to their wild-type littermates (p<0.05; t-test; *Figure 1—figure supplement 1A*). Interestingly, activity levels of wild-type mice during the 12 hr light period did not differ between the two light intensities.

## Expansion of the active period and increased phase advance under DD

After release into DD conditions, $Rai1^{+/-}$ mice maintained a circadian organization of activity and rhythms free-ran with a period length slightly shorter than 24 hr as in wild-type mice (23.8 ± 0.03 hr for both genotypes; p=0.82; t-test). In addition, no significant difference in the robustness of the circadian distribution of locomotor activity was observed during free-run (maximum values of the $X^2$-statistic; $Rai1^{+/+}$ *vs.* $Rai1^{+/-}$: 1458 ± 242 *vs.* 976 ± 84; p=0.086; t-test), while analyses of the day-to-day variability of activity onset under these conditions (i.e., individual *r*-values for the linear regression of activity onset under DD) suggest a more precise circadian timing of overt behaviors in $Rai1^{+/-}$ compared to wild-type littermates (r = 0.96 ± 0.01 and 0.88 ± 0.03, respectively; p<0.05; t-test; see Materials and methods for details). Also, the phase angle of entrainment, which quantifies the relationship between the endogenously generated circadian timing of activity and the timing of light exposure (see Materials and methods for details), differed between the two genotypes. While activity onset in wild-type mice under DD conditions occurred later than predicted by the mean activity onset under the preceding LD12:12 (−0.49 ± 0.17 hr; p<0.005; paired t-test), $Rai1^{+/-}$ mice exhibited a positive phase angle (+0.36 ± 0.14 hr; p<0.05; paired t-test), which significantly differed from wild-type (p<0.005; t-test). This relative advancement of activity onset in $Rai1^{+/-}$ mice was accompanied by an abrupt, 2.5 hr 'expansion' of the active phase (alpha), while in $Rai1^{+/+}$ mice alpha did not significantly change from that observed under LD12:12 (+18 min; *Figure 1A,E*). The alpha expansion was already present on the first day under DD and was due to both an advance of activity onset (consistent with the positive phase angle of entrainment) and a delay of activity offset compared to the timing of these two events on the preceding day under LD12:12 (*Figure 1A*). These observations indicate that besides the altered light-suppression of activity (or 'masking' as it is referred to in the circadian literature), circadian entrainment by light is also affected in $Rai1^{+/-}$ mice. Even at low intensity (i.e., 0.6 cds/m$^2$ *vs.* the habitual 6.6 cds/m$^2$), light was still able to confine activity to the dark period of an LD12:12 light-dark cycle and alpha did not increase in $Rai1^{+/-}$ mice (*Figure 1—figure supplement 1B*).

To further investigate photic entrainment, we assessed the capacity of single, 1 hr light pulses in shifting the phase of activity onset under DD conditions. Light pulses were given at either CT16 or −22, when C57BL/6 mice show maximum phase delays and advances, respectively (*Schwartz and Zimmerman, 1990*; *Daan and Pittendrigh, 1976*). Although both light pulses resulted in the expected change in activity onset of the free-running activity rhythm, the magnitude of the phase advance after light pulses administered at CT22, was more than two times larger in $Rai1^{+/-}$ mice than in wild-type (*Figure 1F*).

## Increased suppression by light of theta-dominated wakefulness but not of total time-spent-awake

Because one of the hallmarks of SMS is disturbed sleep, we next investigated the role of *Rai1* in sleep-wake regulation by recording EEG and EMG signals, concomitant with locomotor activity under various lighting conditions.

Time course analyses under LD12:12 revealed that *Rai1*$^{+/-}$ and *Rai1*$^{+/+}$ mice showed very similar distributions of sleep and wakefulness over the 24 h day (*Figure 2A*; see also Figure 6A) and for none of the three sleep-wake states (i.e., waking, NREM sleep, and REM sleep) did the 24 hr values significantly differ (*Figure 2—source data 1*). Nonetheless, significant differences were observed between genotypes in the sleep-wake distribution. *Rai1*$^{+/-}$ mice displayed more wakefulness (and less NREM sleep) immediately after dark onset and before light onset. Moreover, during the light period, time-spent-awake was slightly but significantly reduced compared to wild-type littermates (*Figure 2B*). Even so, the lower overall activity levels as well as the near complete suppression of locomotor activity by light observed in *Rai1*$^{+/-}$ mice, were not accompanied by similar changes in time-spent-awake. This dissociation resulted in a striking 15-fold reduction in locomotor activity when expressed per unit of time-spent-awake during the light period (*Figure 2C*). Behavioral observation using video recordings confirmed that when *Rai1*$^{+/-}$ mice were behaviorally awake during the light period they did not leave the nest or only very briefly to drink. Because overall activity levels, but not waking (*Figure 2B*), were reduced also in the dark period, activity expressed relative to time-spent-awake was also reduced during this period albeit to a much smaller degree (twofold; *Figure 2C*). Thus, *Rai1*-haploinsufficiency, while suppressing gross body movements during light exposure, leaves general wakefulness largely unaffected.

We then verified whether the immobility in *Rai1*$^{+/-}$ mice was accompanied by altered brain activity during wakefulness. Quantitative analyses of the waking EEG revealed increased activity within the delta band (2–3 Hz) in *Rai1*$^{+/-}$ mice compared to *Rai1*$^{+/+}$ mice during the light period but not during the dark period (*Figure 3A*). This contrast between light and dark could result from differences in the relative contribution of distinct waking behaviors to the overall waking spectra, specifically theta-dominated wakefulness (TDW; [*Welsh et al., 1985*; *Vassalli and Franken, 2017*]) which is considered an active state of wakefulness associated with exploratory behavior and locomotor activity (*Buzsáki, 2002*) (see Materials and methods for the quantification of TDW). Accordingly, time spent in TDW in *Rai1*$^{+/-}$ mice was reduced 2.3-fold in the light period compared to that in wild-type mice (*Figure 2B*), also relative to time-spent-awake (1.9-fold; *Figure 2C*). Thus, in the presence of light, the lack of one *Rai1* allele specifically suppresses active wake behaviors, while leaving total time-spent-awake unaffected.

The EEG spectra during TDW showed the expected theta peak and no differences between genotypes, while for the EEG spectra in the remainder of wakefulness ('quiet' wakefulness or 'non-TDW'), we again observed an increase in EEG power density in the 2–3 Hz range (*Figure 3A*). This indicates that the reduced contribution of TDW to total wakefulness in *Rai1*$^{+/-}$ mice did not underlie this EEG difference. Moreover, the observed increase in the 2–3 Hz range of the EEG power density appeared strictly light driven and not a consequence of for example circadian time, because it was equally observed under LD1:1 conditions when light is presented at all circadian phases (see below and *Figure 3C*). The increase in delta activity in the waking EEG might suggest that when in non-TDW and exposed to light, *Rai1*$^{+/-}$ mice are drowsier. Delta activity in the waking EEG did not only differ between genotypes but was generally higher in the 12 hr light period than in the dark period (2–5 Hz), irrespective of waking state (wakefulness, TDW, and non-TDW) and genotype (*Figure 3B*). However, only in non-TDW was this light-dark delta power difference larger in *Rai1*$^{+/-}$ mice than in *Rai1*$^{+/+}$ mice (*Figure 3B*), suggesting that *Rai1*$^{+/-}$ mice are also more sensitive to light at the level of the EEG. This analysis also revealed that EEG activity in the high-beta / low-gamma activity range (25–40 Hz) was increased in the light versus the dark period and that this increase was again larger in *Rai1*$^{+/-}$ mice (*Figure 3B*).

## Sustained and acute suppression of active wake behaviors in *Rai1*$^{+/-}$ mice

We next assessed whether the suppressing effect of light on locomotor activity and TDW could be sustained beyond 12 hr. When *Rai1*$^{+/-}$ mice were submitted to 60 hr of constant light (LL for 48 hr

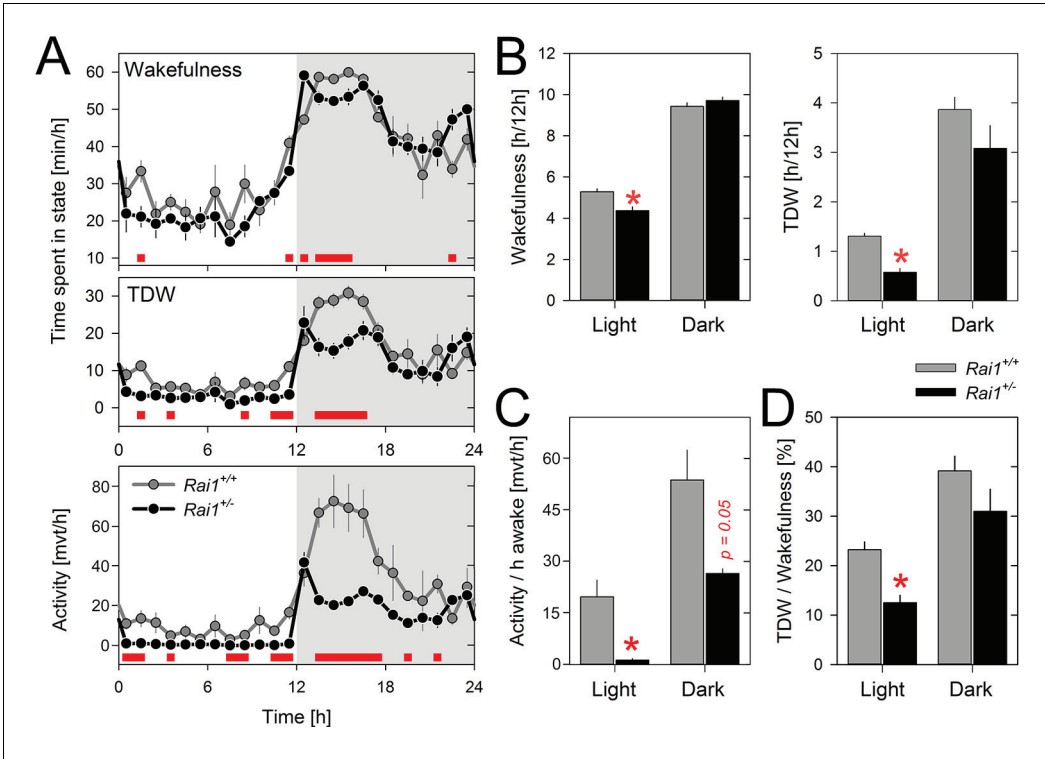

**Figure 2.** Theta-dominated wakefulness (TDW), but not total time-spent-awake, is profoundly suppressed by light in *Rai1*+/- mice. (**A**) Average profile of mean 1 hr values of wakefulness (total wakefulness including TDW), TDW (middle), and locomotor activity (bottom panels) under LD12:12 conditions (n = 4/genotype). Values of two 24 hr baseline days were averaged. Although *Rai1*+/- mice are awake longer than their littermates immediately after dark onset and before light onset, the two genotypes did not differ in time-spent-awake over 24 hr. TDW and activity levels were, however, generally below those observed in wild-type mice (red bars; p<0.05; t-tests). Note that while *Rai1*+/- mice did not show any evidence of dark anticipation in TDW and locomotor activity, we did observe this for wakefulness. In wild-type, levels of wakefulness, TDW, and activity all increase in the hour preceding dark onset. (**B**) Although 24 hr values of total time-spent-awake did not differ between genotypes, we did observe a slight, but significant reduction in *Rai1*+/- mice during the 12 hr light period (*p<0.05, t-test; left panel). Similarly, TDW levels in the light were reduced (*p<0.001; t-test, right panel) but unlike for wakefulness, we did observe an overall reduction in TDW (p<0.05; t-test), compared to wild type. (**C**) Number of movements expressed per hour of total wakefulness were profoundly reduced in *Rai1*+/- mice, especially during the light phase (*p<0.001; t-test) and (**D**) this result was paralleled by a reduction in TDW expressed as a percentage of total wakefulness during the light period, in *Rai1*+/- mice (*p<0.05; t-test). Error bars represent SEMs.

The following source data is available for figure 2:

**Source data 1.** Twenty-four hour values for total wakefulness, theta-dominated wakefulness (TDW), NREM sleep, and REM sleep under LD12:12 conditions.

**Source data 2.** Excel file with data sheets corresponding to each figure panel containing the numerical values on which the *Figure 2* graphs are based.

starting at the end of a 'habitual' 12 hr light period), mean activity levels remained continuously suppressed without marked circadian modulation (*Figure 4A*). Overall activity was reduced to one-fourth of the 24 hr levels under LD12:12 (*Figure 4B*). This is in stark contrast to wild-type mice in which the circadian modulation of locomotor activity was preserved under LL and overall levels did not significantly differ from LD12:12 (*Figure 4A,B*). Interestingly, in neither genotype was the overall time-spent-awake changed under LL (*Figure 4B*). The sustained and important reduction in locomotor activity while leaving wakefulness largely unaffected again attests to the clear dissociation between these behaviors when one *Rai1* allele is missing. Although total wakefulness was not

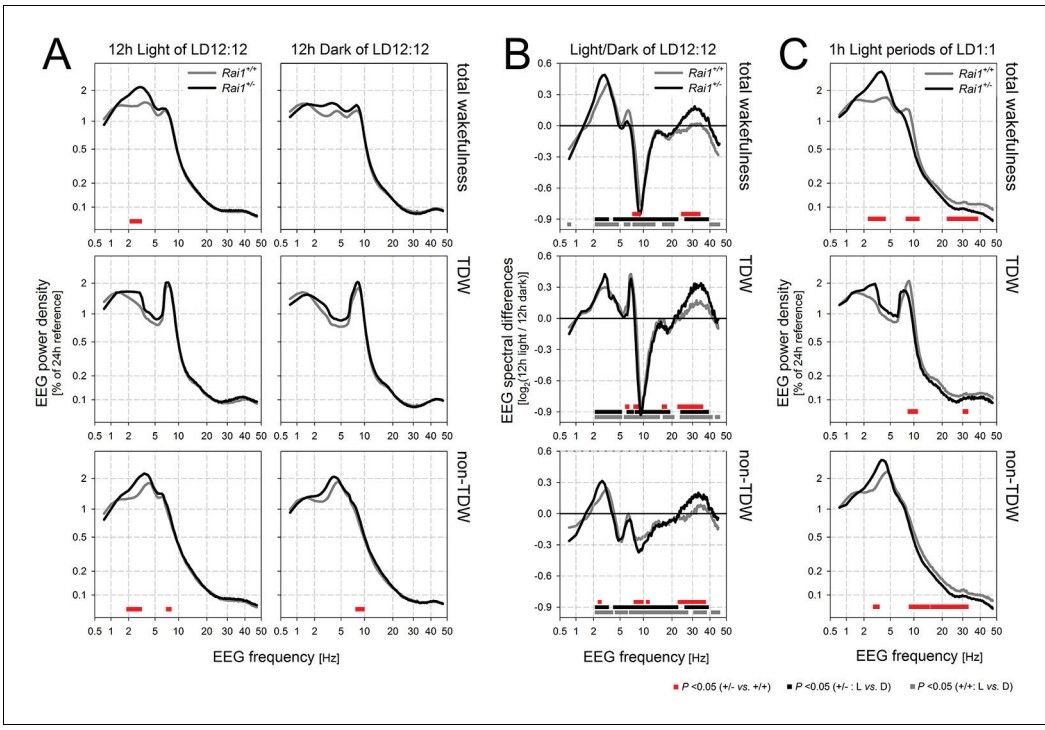

**Figure 3.** Spectral analysis of the waking EEG of $Rai1^{+/-}$ mice show an increased activity within the high delta band. (A) Mean waking EEG spectra during total wakefulness (top), theta dominated wakefulness (TDW; middle), and the remaining wakefulness (non-TDW; lower panels) under the 12 hr light (left) and 12 hr dark (right panels) portion of two baseline days. $Rai1^{+/-}$: black lines and symbols (n = 7); $Rai1^{+/+}$: grey lines and symbols (n = 8). Red bars; p<0.05; t-tests. While the TDW EEG did not differ, in non-TDW ('quiet' waking) power density in the 2–3 Hz range was increased in $Rai1^{+/-}$ mice. (B) EEG spectral light/dark ratios for total wakefulness (top), TDW (middle), and non-TDW (lower panels). Zero levels in each panel represent the EEG activity during the 12 hr dark periods. Both genotypes show light-dark differences in EEG spectral profiles over a large frequency range ($Rai1^{+/-}$: black bars; $Rai1^{+/+}$: grey bars; p<0.05; t-tests). These light-dark differences were modulated by genotype only within the high-delta band and for frequencies between 25–40 Hz (red bars; p<0.05; t-tests). Note that both the prominent downward deflection, maximal close to 10 Hz, and the smaller upward deflection, peaking at around 7 Hz, observed in both genotypes are entirely due to a ~ 0.75 Hz faster TDW theta peak frequency in the dark compared to the light period. (C) The analysis under (A) was repeated for the 24 1h-light pulses under the 48 hr LD1:1 protocol. Again, non-TDW EEG activity in 2–3 Hz range increased when $Rai1^{+/-}$ mice were exposed to light arguing against circadian factors contributing to the results under (A).

affected by LL, TDW was reduced by one third in *Rai1* heterozygous mice compared to LD12:12, and unchanged in their wild-type littermates (*Figure 4A,B*). Although under LL a clear circadian modulation in wakefulness, TDW, and (in wild-type mice only) locomotor activity could still be discerned, the onset of the first wake/active phase was delayed more relative to that under LD12:12 in $Rai1^{+/-}$ mice compared to wild-type (6.9 ± 0.7 *vs.* 2.2 ± 0.5 hr, respectively; p<0.0001, t-test).

After having established that light suppression of active wake behaviors can be sustained for at least 60 hr in $Rai1^{+/-}$ mice, we assessed the acute effects of light. To further investigate the acute effects of light identified under LD12:12 (*Figure 1C*), we implemented a 48 hr LD1:1 light-dark protocol which, in addition, allowed us to determine whether light suppression of activity was modulated by circadian time. In $Rai1^{+/-}$ mice, activity levels were rapidly suppressed and reached near zero levels during each of the repeated 1-hr-light pulses, irrespective of time-of-day (*Figure 5A,B*), giving the changes in activity a distinct 'on-off' or 'switch-like' character (*Figure 5A*, *Figure 5—figure supplement 1*). In contrast, while wild-type mice were also less active during the light pulses, their activity levels remained well above zero and still varied according to circadian time with highest values reached in the subjective dark phase (*Figure 5A,B*). Indeed, in contrast to wild-type mice, activity levels reached during the light pulses in $Rai1^{+/-}$ mice were, on average, similar to the activity

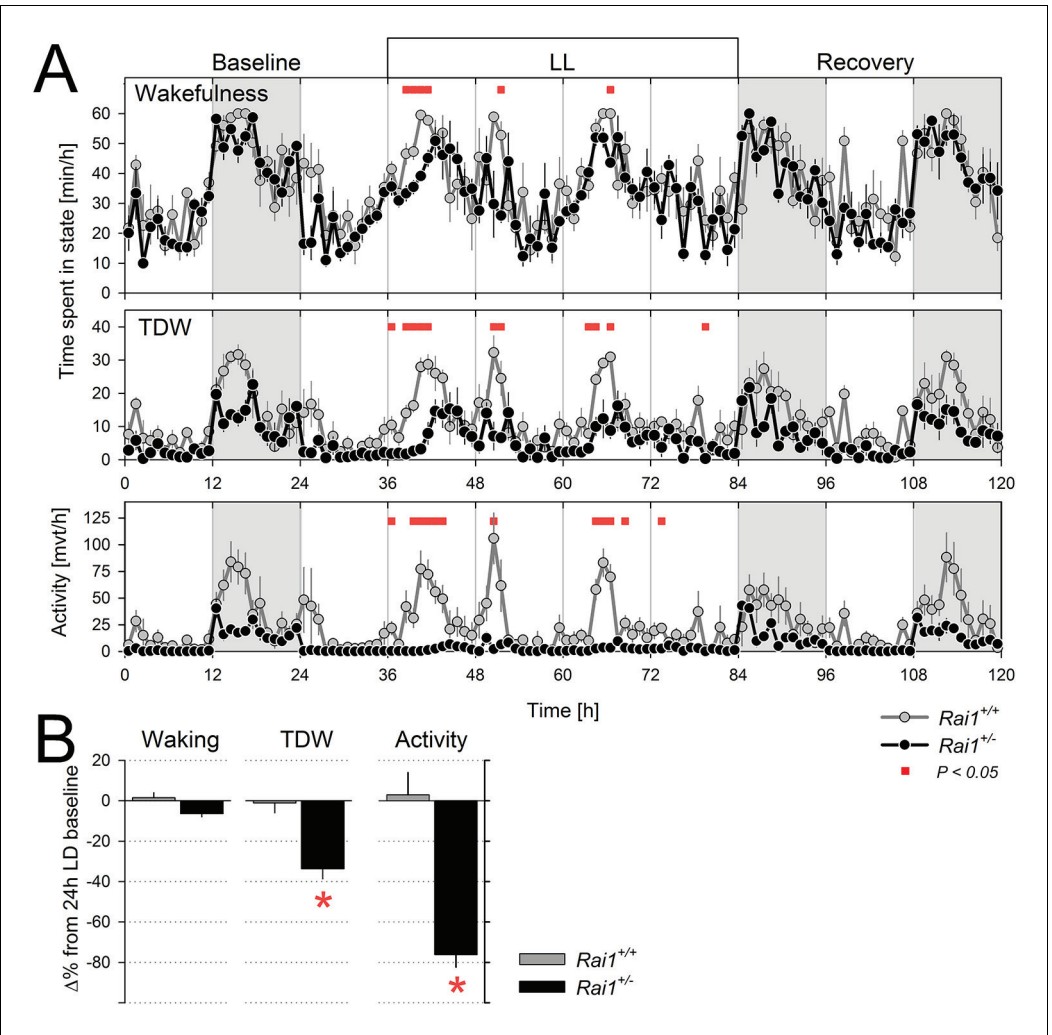

**Figure 4.** Sustained light suppression of locomotor activity and TDW, but not total time-spent-awake, in *Rai1*+/- mice under constant light (LL) conditions. (**A**) Average profiles of mean hourly values of wakefulness (top), TDW (middle), and locomotor activity (bottom panels), under 48 hr LL in *Rai1*+/+ (grey lines and symbols) and *Rai1*+/- (black lines and symbols) mice (n = 4/genotype). In *Rai1*+/- mice LL resulted in a sustained and profound suppression of activity and TDW, leaving wakefulness largely unaffected (red bars; p<0.05; t-tests). Statistical analysis is shown only for the LL condition (for LD12:12 condition see *Figure 2A*) and not for baseline and recovery. (**B**) Wakefulness (left), TDW (middle), and locomotor activity (right panels) expressed as % difference from their 24 hr levels reached under the preceding LD12:12 (=0%). In wild-type mice no suppression of any of the three variables was observed while in *Rai1*+/- mice LL greatly reduced both time spent in TDW and locomotor activity but not total wakefulness (*p<*0.005; t-test*). Errors bars represent SEMs.
The following source data is available for figure 4:

**Source data 1.** Excel file with data sheets corresponding to figure panels A and B containing the numerical values on which the *Figure 4* graphs are based.

(or lack thereof) they exhibited during the light phase under LD12:12 (*Figure 5—figure supplement 2*).

During 1-hr-dark pulses, activity levels varied according to circadian time in both genotypes, but were generally lower in *Rai1*+/- mice than in *Rai1*+/+ mice (*Figure 5A*), consistent with the overall lower activity levels observed during LD12:12. When expressed relative to each animal's mean activity level observed under LD12:12 conditions, *Rai1*+/- mice reached higher values during the 1-hr-dark pulses compared to wild-type (*Figure 5B*), suggesting that the acute activating effect of a dark pulse

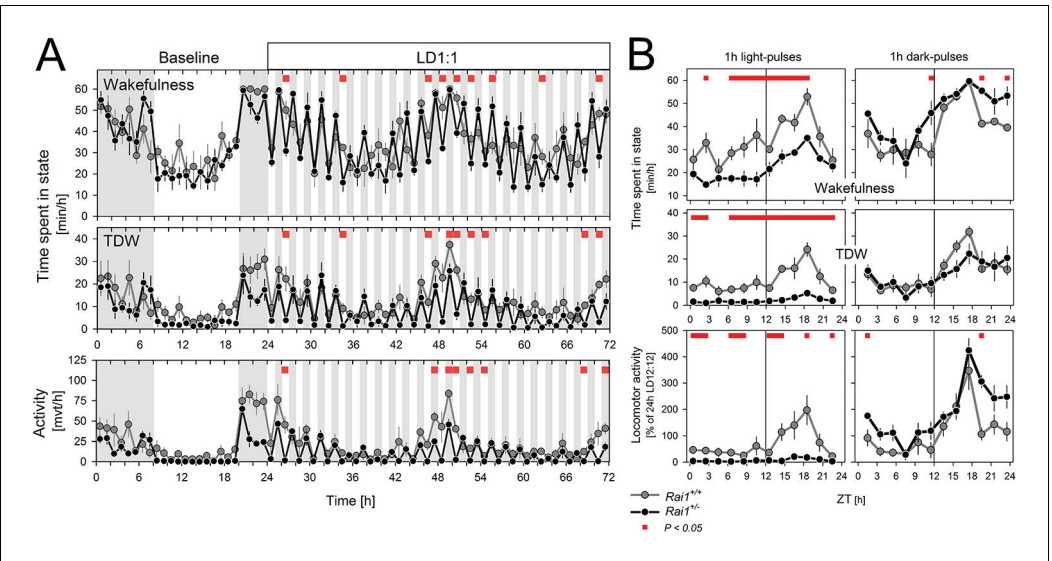

**Figure 5.** Light pulses acutely suppressed locomotor activity and TDW in *Rai1*$^{+/-}$ mice independent of circadian phase. (**A**) Average profile of mean hourly values of total wakefulness (top), TDW (middle), and locomotor activity (bottom panels) under 48 hr LD1:1 in *Rai1*$^{+/+}$ (grey lines and symbols) and *Rai1*$^{+/-}$ (black lines and symbols) mice (red bars; p<0.05; t-tests; n = 4/genotype). Note the 'saw-tooth' pattern for all three variables in *Rai1*$^{+/-}$ mice, with TDW and activity reaching near 0 levels during the 1-hr-light pulses. (**B**) Wakefulness (top), TDW (middle), and locomotor activity (bottom panels) during the 1-hr-light (left) and 1-hr-dark pulses (right panels) averaged for the 2 days under LD1:1. Relative activity is calculated as % of the individual's total activity under the preceding day under LD12:12. Levels of activity and TDW during the 1-hr-light pulses follow a circadian pattern in *Rai1*$^{+/+}$ (grey lines and symbols), but not in *Rai1*$^{+/-}$ (black lines and symbols) mice. Under LD1:1 also wakefulness is suppressed to a higher degree in *Rai1*$^{+/-}$ mice compared to wild-type (red bars; p<0.05; t-tests). Wakefulness, TDW, and activity levels during the 1h-dark pulses were less affected by genotype, although wakefulness and relative activity levels were generally higher in *Rai1*$^{+/-}$ mice than wild-type mice, indicating increased acute activation by darkness.

The following source data and figure supplements are available for figure 5:

**Source data 1.** Excel file with data sheets corresponding to figure panels A and B containing the numerical values on which the *Figure 4* graphs are based.

**Figure supplement 1.** Illustration of the acute and sustained suppression of activity by light in *Rai1*$^{+/-}$ mice under different lighting regimens (LD12:12, LD1:1, and LL).

**Figure supplement 2.** Summary of locomotor activity differences during light exposure for the three lighting conditions.

was also amplified in *Rai1*$^{+/-}$ mice, which is consistent with the pronounced increase in activity at the light-to-dark transitions under LD12:12 (*Figure 1C*).

Effects of light, time-of-day, and genotype on TDW largely paralleled the effects on activity, although values did not differ between genotypes during the dark pulses (*Figure 5A,B*). Also, wakefulness was suppressed by the light pulses and this suppression was stronger in *Rai1*$^{+/-}$ mice compared to wild-type (*Figure 5B*). This markedly differs from the lack of suppression of wakefulness by light under LD12:12 and LL, demonstrating that the direct effects of light differ between acute and sustained light administration. Evidence of a stronger, acute activation by the dark pulses was also observed for wakefulness (*Figure 5B*), which in combination with the stronger acute suppression by light resulted in a larger amplitude of change in *Rai*$^{+/-}$ mice between light and dark pulses, resulting in a distinct saw-tooth pattern (*Figure 5A*).

## Rai1 haplo-insufficiency does not affect the sleep homeostat

Next, we verified whether genotype-driven differences in the homeostatic aspect of sleep regulation could have contributed to the differences in wake behavior. We therefore challenged mice of both genotypes with a 6-hr sleep deprivation during the first half of the light period (ZT0-6) and compared their compensatory sleep responses. Compared to baseline levels, mice of both genotypes spent more time asleep during the 18 hr for which recovery sleep was monitored and this extra time asleep did not differ between genotypes (*Figure 6A,B*). Similarly, the dynamics of EEG delta power over the 72 hr recording period, including its levels immediately after the sleep deprivation, did not differ between genotypes (*Figure 6A*). EEG delta power during NREM sleep quantifies the amplitude and prevalence of delta oscillations (1–4 Hz) and is thought to reflect a sleep homeostatic process in both humans and mice (*Daan et al., 1984*; *Franken et al., 2001*). In sum, at least in the mouse, altered sleep homeostasis does not seem to contribute to the sleep-wake phenotypes associated with *Rai1* haplo-insufficiency.

## Rai1 expression in the forebrain does not vary with time-of-day

*Rai1* is widely expressed throughout the brain with high levels of expression observed in the olfactory bulb, cerebral cortex, hippocampus, striatum, thalamus, hypothalamus, cerebellum, and brainstem (*Toulouse et al., 2003*; *Bi et al., 2007*; *Fragoso et al., 2015*; *Huang et al., 2016*). Because circadian and/or light-dependent changes in *Rai1* expression in the brain may contribute to the phenotypes we observed in *Rai1*-haploinsufficient mice, we sampled brain tissue under LD12:12 conditions at 3-hr intervals in wild-type mice. *Rai1* expression levels in the forebrain were remarkably stable over the day, in contrast to the well-known nychthemeral changes in steady state mRNA levels

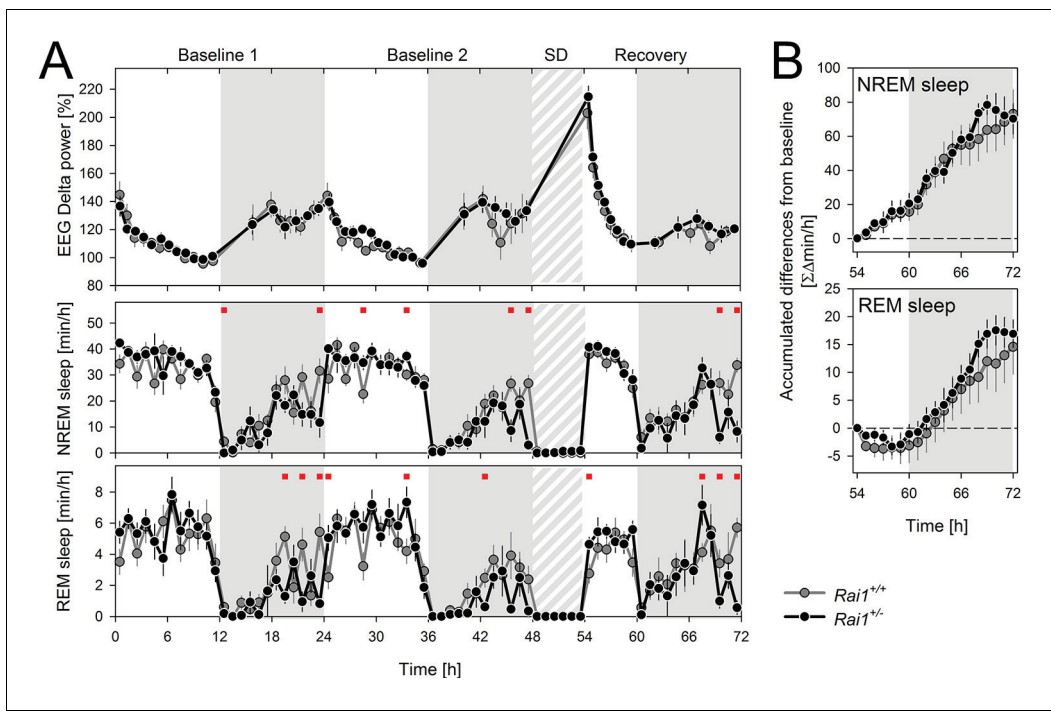

**Figure 6.** Sleep homeostasis is not altered in *Rai1*[+/-] mice. (**A**) Average time course of EEG delta power (1–4 Hz) in NREM sleep (top), time spent in NREM sleep (middle) and in REM sleep (bottom panel) under LD12:12. After 2 baseline days (0–48 hr), animals were sleep deprived for 6 hr (48–54 hr) by gentle handling starting at light onset. Recovery was monitored over the remaining 18 hr of the 3-day recording (54–72 hr). *Rai1*[+/+]: grey lines and symbols; *Rai1*[+/-]: black lines and symbols. Red squares above data mark significant genotype differences (p<0.05; t-tests). (**B**) Mean hourly accumulation of differences from baseline in time spent in NREM (top) and REM (bottom panel) sleep during the 18 hr of recovery sleep after the SD (54–72 hr) did not differ between genotypes. *Rai1*[+/-] n = 7; *Rai1*[+/+] n = 8.

of the established core circadian clock genes *Bmal1* and *Per2*, which were quantified in the same samples (*Figure 7*). A database search confirmed that *Rai1* is present also in the SCN and that its expression does not change with time-of-day (CircaDB; http://circadb.hogeneschlab.org/).

## Decreased retinal and SCN response and increased vSPVZ response to light in *Rai1⁺/⁻* mice

In a further step to dissect the pathways by which locomotor activity was so strongly suppressed in *Rai1⁺/⁻* mice, we quantified the retinal response to light pulses in dark-adapted mice. Scotopic ERG recordings revealed that retinal activation was importantly reduced, as evidenced by a decreased amplitude of the *a*-wave, reflecting the membrane hyperpolarization of the photoreceptors (*Penn and Hagins, 1969*), and the *b*-wave , reflecting the light response of retinal cells post-synaptic to photoreceptors (*Stockton and Slaughter, 1989*; *Miller and Dowling, 1970*) (*Figure 8A,B*). Together, these two effects led to a marked flattening of the ERG response in *Rai1⁺/⁻* mice. Since the *b*-wave is a consequence of the activity of the components measured by the *a*-wave (photoreceptors), we cannot conclude whether the decreased *b*-wave is exclusively due to altered photoreceptor function or also to intrinsic interneuron defects.

The general retinal morphology and histology analysis did not show any salient differences between *Rai1⁺/⁺* and *Rai1⁺/⁻* mice (*Figure 9*). No decrease in layer thickness of photoreceptors or interneurons was noticed that might have been indicative of retinal degeneration explaining the reduced ERG response to light. The lack of a difference in the rod marker rhodopsin, the cone marker GNAT2, the bipolar ON marker PKCalpha, and the active Müller glial marker GFAP, further supported that retinal composition was not noticeably impacted (*Figure 9A–H*). Moreover, the

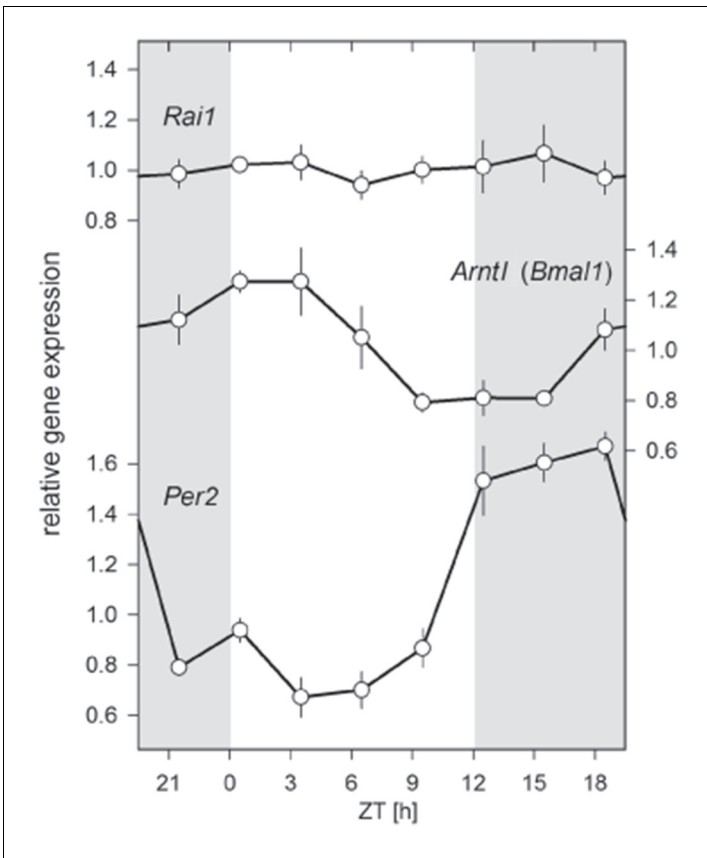

**Figure 7.** Twenty-four hour brain expression profiles for *Rai1*, *Arntl*, (*Bmal1*), and *Per2* in wild-type mice. While *Bmal1* and *Per2* expression showed their well-known anti-phasic circadian expression profiles, *Rai1* expression was remarkable indifferent to time-of-day and lighting condition. Samples were taken at 3 hr intervals (total n = 23; that is, n = 3/interval, except for ZT18: n = 2).

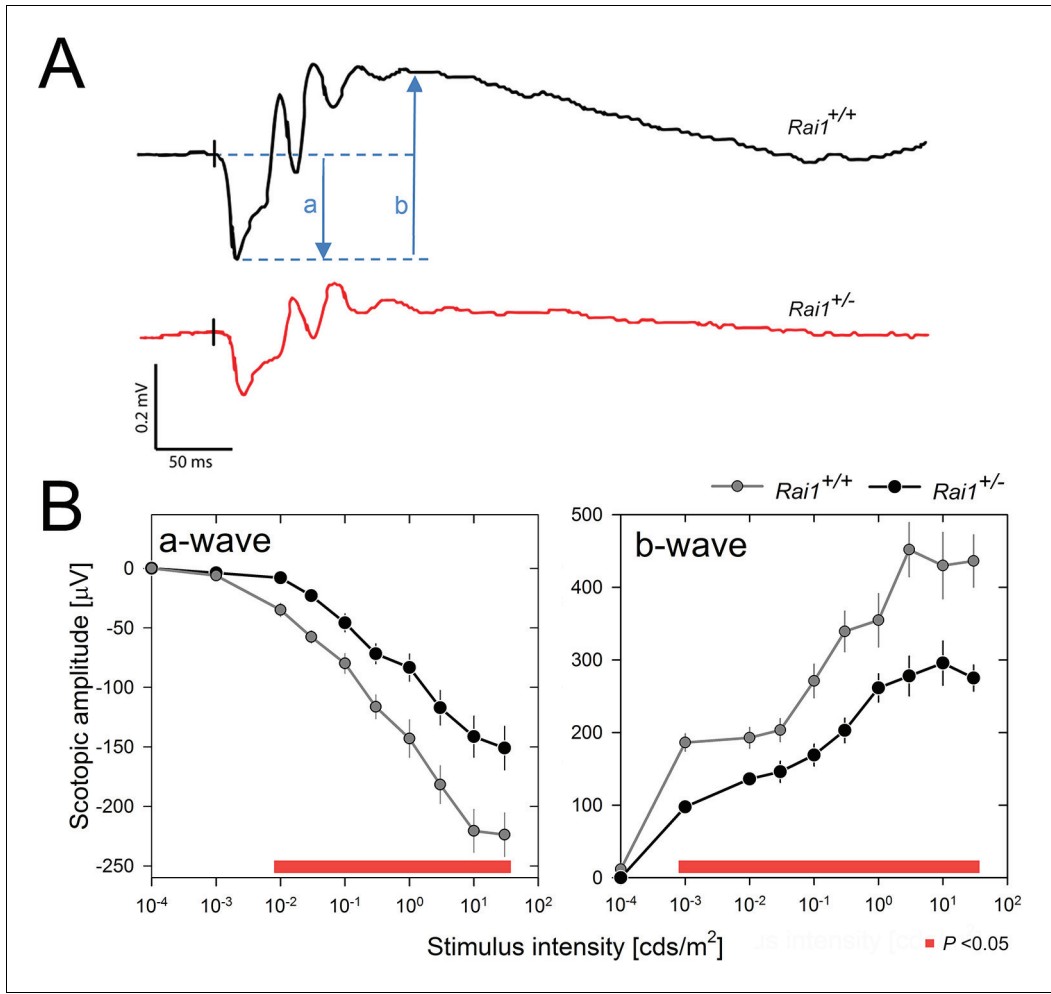

**Figure 8.** Decreased light-induced retinal activity in *Rai1* haplo-insufficient mice. (**A**) Representative example of ERG responses of a *Rai1*[+/+] mice (upper panel; black line) and a *Rai1*[+/-] mice (lower panel; red line) to single flashes of 30 cds/m[2] recorded under scotopic conditions. The letters 'a' and 'b' represent *a*- and *b*-wave amplitudes, respectively. Vertical black tick marks just prior the *a*-wave indicate timing of the light stimuli. Scale bars, horizontal: 50 ms, vertical: 0.2 mV. (**B**) *Rai1*[+/-] mice (black symbols and lines) displayed a significantly smaller ERG response compared to *Rai1*[+/+] mice (grey symbols and lines), due to both a smaller (less negative) *a*-wave (left) and *b*-wave (right panels) amplitude over a broad range of stimulus intensities (red bars; p<0.05; t-test).

The following source data is available for figure 8:

**Source data 1.** Excel file with two data sheets corresponding to *Figure 8* panel B a-wave and b-wave values, respectively.

---

normal dendritic ends of *Rai1*[+/-] bipolar cells, labeled by PKCalpha, suggest that the connections between photoreceptors and bipolar cells were not affected, as well as the general circuitry as shown by the calbindin, ChAT, and GABA labeling of the horizontal and/or amacrine cells and the different strata of connections in the inner plexiform layer (*Figure 9I–N*). Consistent with the histological results, qPCR analyses of retinal extracts did not reveal any difference in *rhodopsin*, *melanopsin*, and *cone opsin* mRNA levels between *Rai*[+/-] and *Rai*[+/+] mice (*Figure 9O*). We did, however, observe a twofold decrease in *Rai1* expression in *Rai1*[+/-] mice, as could be expected from the haplo-insufficiency (*Figure 9O*). Taken together, these experiments did not reveal any anatomical or molecular factors contributing to the reduced ERG response.

Next, we verified whether a light pulse in *Rai1*[+/-] mice is also less efficient in activating the SCN by quantifying c-FOS immunoreactivity. In line with previous work (e.g. [*Ruby et al., 2002*]), a 1-hr-

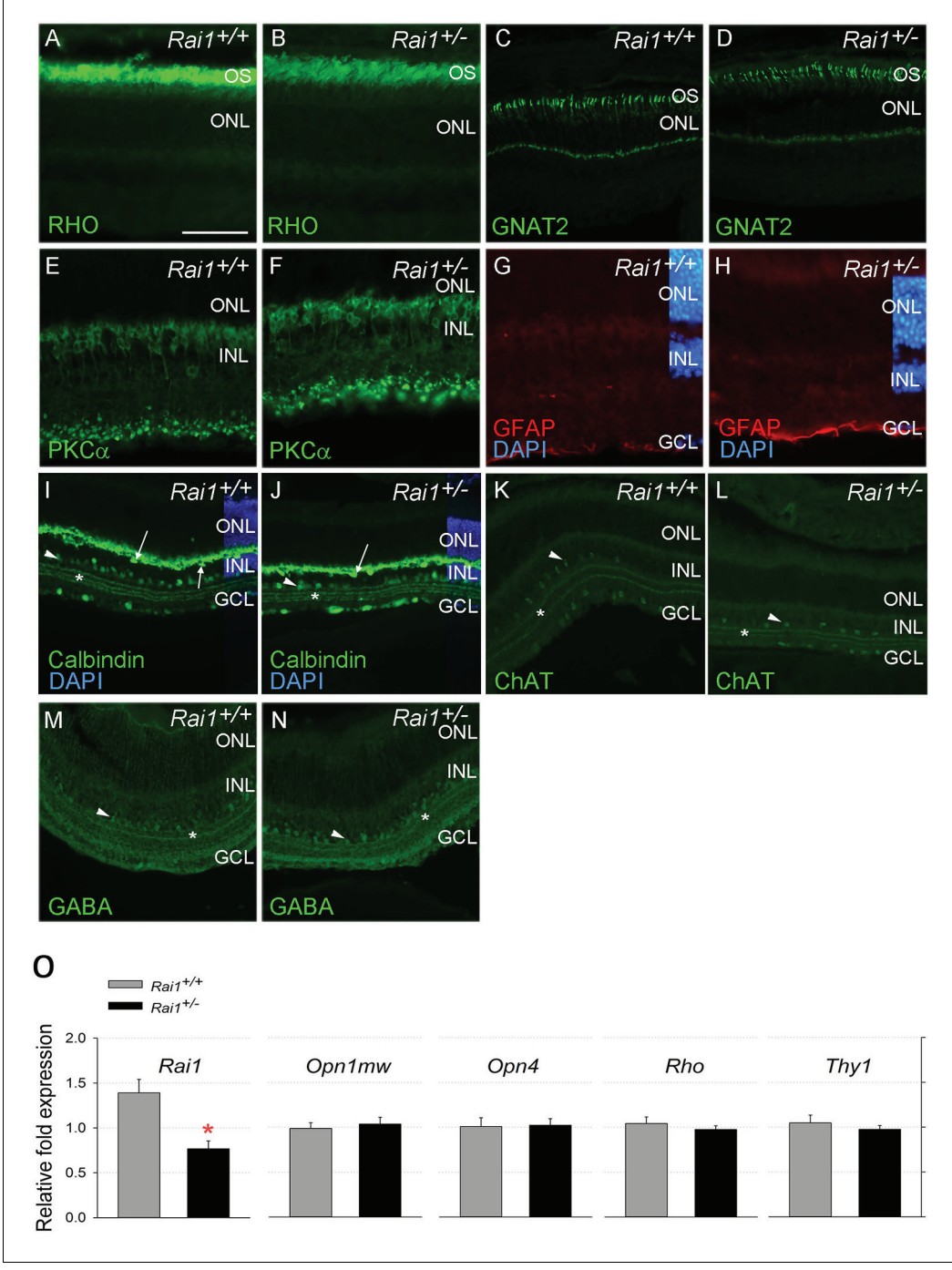

**Figure 9.** Retinal protein and gene expression levels were normal in *Rai1*$^{+/-}$ mice. Immunostaining for Rhodopsin protein (**A, B**), the cone protein GNAT2 (**C, D**), the bipolar protein PKCa (**E, F**), the Müller cell protein GFAP (**G, H**), and Calbindin (**I, J**), ChAT (**K, L**), and GABA (**M, N**) markers did not differ between *Rai1*$^{+/-}$ (**B, D, F, H, J, L, N**) and *Rai1*$^{+/+}$ mice (**A, C, E, G, I, K, M**). As expected, rhodopsin was mainly localized in outer-segments of rods (**A, B**) and GNAT2 in cone cells (**C, D**). PKCa labeled whole ON bipolar cells with both dendrite and synapse structure (**E, F**). GFAP expression was restricted to the foot of Müller cells not extending to their cell bodies that show no activation in these retinal cells (**G, H**). Calbindin was expressed in horizontal cells (arrows) and some amacrine cells (arrowhead) as well as in ganglion cells located in the GCL (**I, J**). Three strata of connections (star) were well labeled by calbindin in both genotypes in the layer connecting the INL and GCL layers. Both ChAT (**K, L**) and GABA (**M, N**) labeled subpopulations of amacrine cells (arrowheads) and different strata of connections (star) in a pattern similar for *Rai1*$^{+/-}$ (**L, N**) and *Rai1*$^{+/+}$ (**K, M**) mice. White horizontal bars represents 50 μm in A, B, E, F, G,

*Figure 9 continued on next page*

*Figure 9 continued*

and H and 100 μm in C, D, and I through M. OS: outer-segments; ONL: outer-nuclear layer; INL: inner-nuclear layer; GCL: ganglion cell layer; RHO: rhodopsin in green; GNAT2: G-protein subunit alpha transducin two in green; PKCa: Protein kinase C alpha in green; GFAP: gGlial fibrillary acidic protein in red; DAPI: 4′,6-diamidino-2-phenylindole in blue; ChAT: choline acetyltransferase ChAT in green; GABA: γ-aminobutyric acid in green. (**O**) Quantitative RT-PCR showed that expression of M-opsin (*Opn1mw*; cones), melanopsin (*Opn4*, ipRGCs), *rhodopsin* (*Rho*; rods), and the retinal ganglion cell marker *Thy1* was normal in *Rai1*$^{+/-}$ mice. *Rai1* expression was decreased by 50% in *Rai1*$^{+/-}$ mice, as could have been expected.

light pulse administered at ZT15-induced c-FOS immunoreactivity in the SCN of both genotypes (*Figure 10B,C*; $p<0.0001$, *t-tests*). The induction was, however, significantly reduced by ~40% in *Rai1*$^{+/-}$ mice (*Figure 10B,C*; $p<0.005$, t-test). Although the decreased retinal activation upon a light pulse is consistent with the reduced downstream response in the SCN, both observations seem to contradict the acute and sustained behavioral hypersensitivity to light we observed in *Rai1*$^{+/-}$ mice. This profound reduction in the activation of the SCN seems also at odds with the lack of a difference between the two genotypes in the ensuing phase delay of overt behavior after a light pulse (*Figure 1F*). However, at the level of the ventral subparaventricular zone (vSPVZ), a region immediately dorsal to the SCN, the opposite result was observed. The SPVZ extends dorsally of the SCN, from bregma −0.71 mm to bregma −0.83 mm (*Paxinos and Franklin, 2013*) and we focused on its ventral part (*Figure 10A*) because neuronal activity in this area was found to be in phase with locomotor activity rhythm (*Todd et al., 2012*) and because stimulation of this area inhibits locomotor activity specifically (*Kramer et al., 2001*). We found that compared to mice that did not receive a light pulse, light induced a significant increase in c-FOS immunoreactivity only in *Rai1*$^{+/-}$ mice ($p<0.05$; t-test) and that the number of c-FOS-positive cells after the light pulse was higher in *Rai1*$^{+/-}$ mice compared to their wild-type littermates ($p<0.05$, t-test; *Figure 10C*). The differential activation by light of the frontal vSPVZ in *Rai1*$^{+/-}$ mice could thus represent the neuronal substrate underlying light's marked suppression of active waking behaviors (*Figure 11*).

## Discussion

### *Rai1* haplo-insufficiency exacerbates the direct effects of light on behavior

Our data revealed that *Rai1* haplo-insufficiency in the mouse resulted in both circadian and light-induced phenotypes while leaving the homeostatic regulation of sleep unaffected. The most striking effect of the lack of one *Rai1* allele was the all but complete abolishment of locomotor activity by light. The overall time-spent-awake and its diurnal distribution under standard LD12:12 conditions did not greatly differ between the two genotypes, underscoring that locomotor activity in the mouse can be a poor indicator of sleep-wake state.

Although overall wakefulness was little affected, the profound suppression by light of locomotor activity was accompanied by a number of changes in wake-related phenotypes indicating reduced alertness and attention while awake during the light in *Rai1*$^{+/-}$ mice compared to their wild-type littermates. These include a reduction in time spent in TDW, a sub-state of wakefulness rich in EEG theta activity and associated with active, exploratory, and goal-oriented behavioral repertoires (*Buzsáki, 2002*), and an increase in delta activity in the waking EEG during the light. The latter observation suggests that *Rai1*$^{+/-}$ mice are less alert and more drowsy when awake in the light, and is consistent with reports of delta activity increasing progressively during prolonged waking periods in human and rat, which in human is paralleled by decrements in cognitive performance (*Cajochen et al., 2002*; *Franken et al., 1991*; *Dijk et al., 1992*). Also, the high beta/low gamma frequency band was more strongly activated by light in *Rai1*$^{+/-}$ mice than in wild-type. This frequency range also increases progressively during prolonged waking in both human and mouse (*Cajochen et al., 2002*; *Grønli et al., 2016*) and has been associated with behavioral arrest and slower movements (*Zheng et al., 2015*), as well as with the motor deficits accompanying Parkinson disease (*Weinberger et al., 2009*; *Mallet et al., 2008*), suggesting that processes related to motor

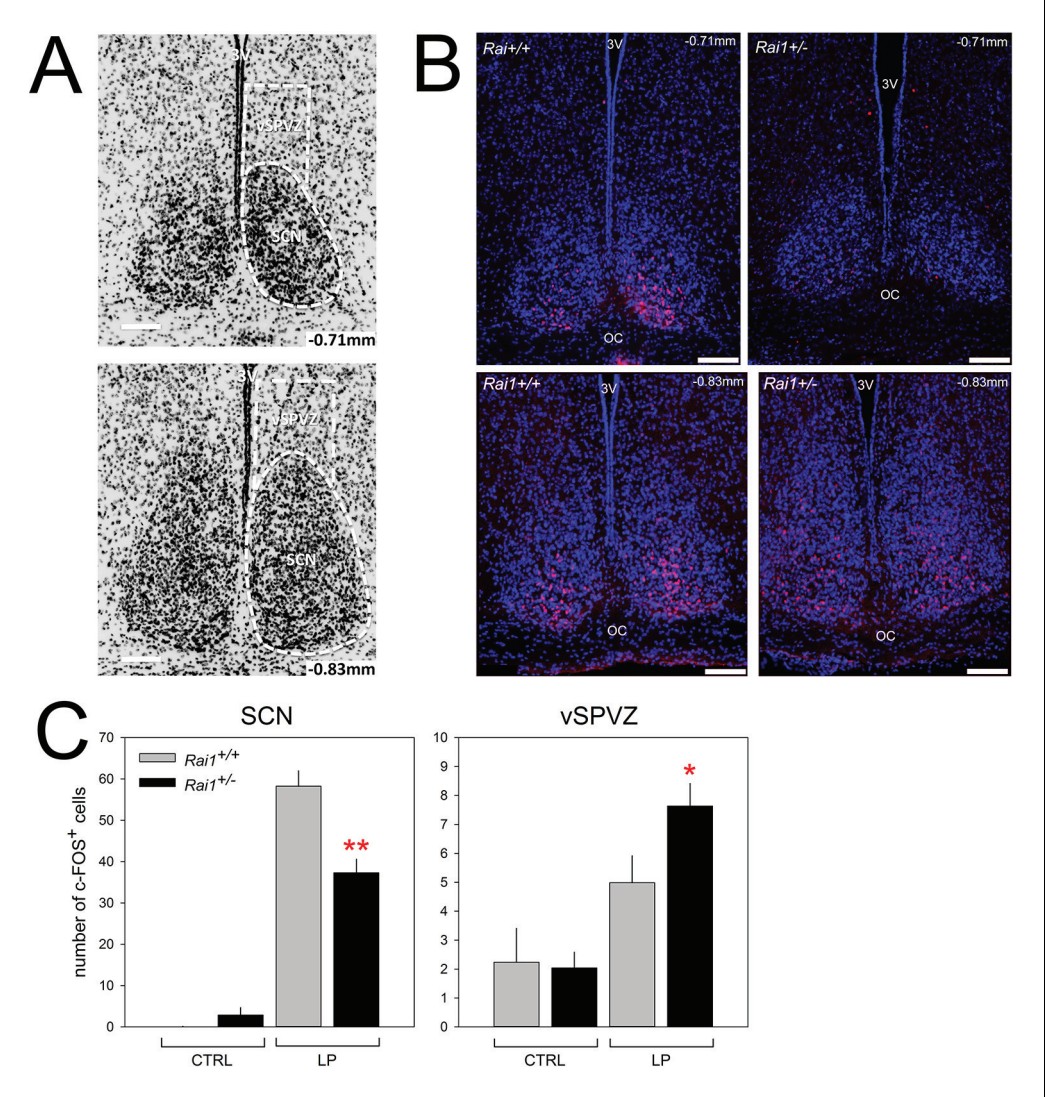

**Figure 10.** Decreased SCN activation and increased vSPVZ activation by light in *Rai1* haplo-insufficient mice. (**A**) Illustration of anatomical delimitation (white dashed lines) of SCN and vSPVZ regions at bregma ~ −0.71 mm and ~ −0.83 mm used to count c-FOS-positive (c-FOS+) cells in B and C. SCN contour is readably visible by the marked density of cells within the SCN. DAPI staining in black labels cell nuclei. The vSPVZ contour was defined following published coordinates (*Paxinos and Franklin, 2013*). Note that top photo in A is identical to top-left photo in B with DAPI in blue on black background allowing visualization of c-FOS immunofluorescence in red in the ventral retinorecipient SCN core (**B**) Example of representative immunofluorescence staining of four sections (upper two panels at −0.71 mm, lower 2–0.83 mm) of two wild-type (left) and two *Rai1+/-* heterozygous (right panels) mice. White horizontal bars in each photo of Panels A and B mark 100 μm. (**C**) A 1-hr-light pulse (LP) produced a smaller c-FOS immunoreactivity response in the SCN of *Rai1+/-* mice compared to wild-type littermates. This is reflected by a reduced number of c-FOS-positive cells (left panel; **p*<0.005; t-tests*). In the vSPVZ, a 1-hr-light pulse produced a larger response in c-FOS immunoreactivity in *Rai1+/-* mice (right panel; **p*<0.05; t-test; n = 6 and 7 in Rai1+/- and Rai1+/+ mice*). Two control groups did not receive a light pulse (CTRL: n = 3/genotype). Note the different scaling for the SCN and vSPVZ panels. c-FOS immunoreactivity was averaged for the −0.71 and −0.83 mm sections within animals.

The following source data is available for figure 10:

**Source data 1.** Excel file with two data sheets corresponding to *Figure 10* panel C SCN and vSPVZ number of cFOS-positive cells, respectively.

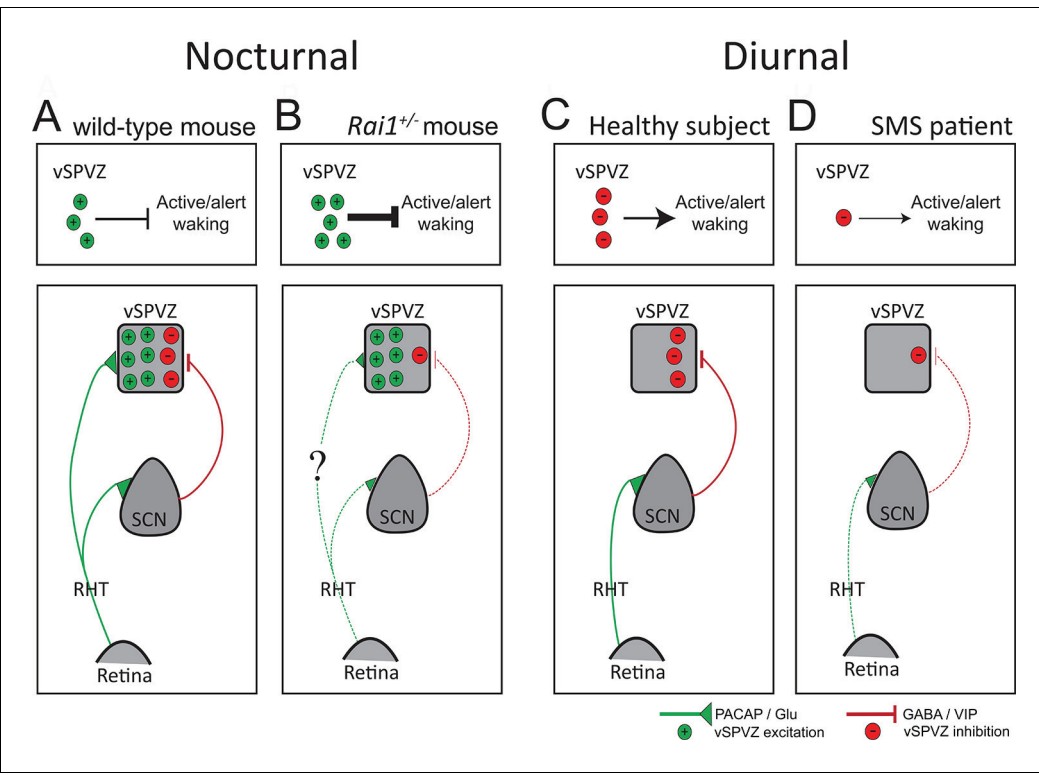

**Figure 11.** Diagram illustrating a hypothesis explaining the differential response to light, from retina to behavior, in SMS mice and patients. (**A**) In the intact, nocturnal mouse light stimulates the retina and, through the PACAP and GLU excitatory projections (green) of the retinohypothalamic tract (RHT), activates both the SCN and the vSPVZ. Upon activation, the SCN, in turn, inhibits the vSPVZ through GABA/VIP inhibitory projections (red). RHT activation outweighs SCN inhibition resulting in a net activation of the vSPVZ causing suppression of active wake behaviors and locomotor activity during a light pulse. The four upper panels illustrate the net excitatory-inhibitory balance in the vSPVZ as number of green (+, excitatory) or red (-, inhibitory) circles and the effect on active/alert wake behaviors. (**B**) In *Rai1*+/- mice, despite a reduced retinal response to light, excitatory inputs received from the RHT are somehow ('?') sufficient to correctly activate the vSPVZ (see *Figure 10C*). However, SCN activation by the RHT is impaired resulting in a weaker inhibition of the vSPVZ (bottom panel). The net effect of normal excitation and impaired inhibition of the vSPVZ is an over-activation explaining the exacerbated suppression of locomotor activity (upper panel). (**C**) In humans, light information encoded in the retina and transmitted by the RHT, activates only the SCN as the retinal-vSPVZ projections are thought not to be functionally important in diurnal species (*Todd et al., 2012*). In turn, the SCN will inhibit the vSPVZ (bottom) thus promoting active wake behaviors (upper panel). (**D**) In SMS patients, as in the mouse model, a reduced retinal response to light might lead to a reduced SCN stimulation and a reduced suppression of the vSPVZ (bottom) and reduced promotion of active/alert wake behaviours during the day (upper panel). PACAP: Pituitary adenylate cyclase-activating polypeptide; GLU: Glutamate; GABA: γ-aminobutyric acid; VIP: vasoactive intestinal peptide; dotted green/red lines: weaker stimulation/inhibition. Inspired by *Todd et al., 2012*).

control are reflected in the waking EEG and might be relevant also for the light-induced suppression of locomotor activity in the SMS model.

Results from the LD12:12, LD1:1, and LL experiments also allowed us to differentiate between sustained and acute direct effects of light on sleep-wake state. In the context of the current set of experiments, we refer to acute effects as those light effects that can be observed immediately after light onset; that is, within the first hour, while sustained refers to effects of longer light exposures; that is, 12 hr and longer. Results from the LD12:12 and LL experiments revealed that genotype did not alter the effects of sustained light exposure on total time-spent-awake (and -asleep), whereas the LD1:1 experiment provided evidence that the acute suppressing effects of light on time-spent-wake time were stronger in *Rai1*+/- mice, as was the case for locomotor activity and TDW. In contrast to the almost complete suppression of locomotor activity and TDW during LD1:1, the effect on total

time-spent-awake was weaker and depended on circadian time, with highest levels of wakefulness attained (or lowest suppression by light) when light pulses were given during the subjective dark phase. This suggests that the sleep-wake switching by light competes with the alerting signal provided by the circadian system, as it is more difficult for light to acutely switch sleep-wake state than to rapidly alter the quality of one state (e.g. active vs. quiet wakefulness).

In addition, the LD1:1 protocol revealed a stronger induction of wakefulness during the 1-hr-dark pulses in $Rai1^{+/-}$ mice compared to wild-type. Also this acute, wake-promoting effect of the absence of light varied with circadian time and was largest at the start of the subjective dark phase. In comparison with the temporary overshoot in wakefulness and activity at the same time under LD12:12, which could be interpreted as 'rebound' from the suppression in the preceding 12 hr of light, the LD1:1 results instead suggests a circadian mechanism that could relate to the longer circadian active period observed in $Rai1^{+/-}$ mice under DD conditions.

### Rai1 haplo-insufficiency and circadian rhythm entrainment

In the present study, circadian organization of locomotor activity under DD was not largely impacted by $Rai1$ haplo-insufficiency, and no differences in period length were observed. Our results contrast with those from one earlier study in which a 13 min shortening of the free-running period was reported (*Lacaria et al., 2013*). Despite this difference, both studies clearly demonstrate that circadian rhythm generation under DD is not compromised in $Rai1^{+/-}$ mice. This is surprising given that $Rai1$ haplo-insufficiency results in the transcriptional dysregulation of core clock genes such as *Clock*, *Per2*, *Per3*, *Cry1*, and *Bmal1* in SMS fibroblasts and in mouse hypothalamic tissue (*Williams et al., 2012*). Such extensive disruption of the core clock machinery could have been expected to impact the circadian organization of overt behaviors in vivo. If anything, our results show that circadian timing under constant dark conditions was more accurate in $Rai1^{+/-}$ mice than in wild-type mice.

We did, however, identify two circadian phenotypes related to entrainment. $Rai1^{+/-}$ mice showed a pronounced lengthening of the active phase (alpha) when released into DD. This expansion of alpha was due to both an earlier onset and a later cessation of activity compared to the timing of these two events under the preceding LD12:12 conditions. Given their hypersensitivity to light, the preferred (i.e. endogenously determined by the circadian system) earlier activity onset in $Rai1^{+/-}$ mice seems thus 'masked' by light even of low intensity. Evidence of masking of activity onset in $Rai1^{+/-}$ mice can also be deduced from the lack of transients during re-entrainment to LD12:12 after a period of DD (*Figure 1A*). The second and related circadian phenotype observed in $Rai1^{+/-}$ mice is the twofold larger phase advance. We did not construct a full 'phase response curve' (PRC) but assuming a still larger phase advance 2 hr later (i.e. at light onset under entrained conditions) might have contributed to the earlier preferred activity onset in $Rai1^{+/-}$ mice and also to the compression of alpha under entrained, LD12:12 conditions (compared to DD). Alpha compression during re-entrainment is accompanied by memory impairments (*Ruby et al., 2015*) and, assuming that in $Rai1^{+/-}$ mice alpha is being compressed by 2.5 hr on a daily basis under LD12:12 conditions, might also have contributed to cognitive deficits reported for $Rai1$-compromised mice kept under these same lighting conditions (*Bi et al., 2007*; *Huang et al., 2016*). It is therefore of interest to verify whether memory deficits persist under LD regimens with shorter than 12 hr photoperiods better matching the longer endogenous active period observed in $Rai1^{+/-}$ mice.

### Reconciling species differences and 'input-output' differences

The sleep-wake disturbances, including excessive daytime sleepiness and attention deficits (*Smith et al., 1986*), together with the inverted melatonin rhythm observed in SMS patients (*De Leersnyder, 2006*; *Potocki et al., 2000*), strongly suggest impaired non-visual light perception. The reduced response to light at the level of the retina and the SCN we report here for $Rai1^{+/-}$ mice, supports this idea and indicates that already at the input level, processing of light information is altered by the loss of one $Rai1$ allele. However, at the behavioral output level, the murine SMS model clearly differs from SMS patients because mice display a striking hypersensitivity to light with respect to active waking behaviors. These findings seem contradictory not only with regard to the human-mouse species difference, but also concerning the differences in the effects of light at the input level and its downstream behavioral consequences in our SMS mouse model. This

contradiction might relate to the mechanisms that determine nocturnality versus diurnality which are thought to act downstream from the SCN.

Based on the neuroanatomical differences in the innervation of the vSPVZ by the SCN and the retina between nocturnal and diurnal rodents, Todd and colleagues *Todd et al., 2012* proposed that in diurnal mammals light information reaches the vSPVZ predominantly through the SCN relay, which inhibits the vSPVZ and is thus alerting. If also true in humans, a weaker inhibition of the vSPVZ by the SCN might contribute to the lower levels of alertness and attention, as well as increased day-time sleepiness observed in SMS during the day (*Figure 11C,D*) (*Elsea and Girirajan, 2008*). In nocturnal species, light information encoded in the retina directly stimulates the vSPVZ via the retinohypothalamic tract (RHT) and, overcoming the simultaneous inhibition provided by the SCN, causes inactivity and sleep (*Todd et al., 2012*; *Schwartz et al., 2004*; *Watts et al., 1987*; *Colwell, 2011*; *Hermes et al., 2009*) (summarized in the *Figure 11* schematic). According to this scenario, the reduced activation of the SCN observed in our SMS mouse model might have led to a reduced inhibition of vSPVZ neurons, which, in turn, explains both the stronger induction of vSPVZ c-FOS immunoreactivity we observed, as well as the exacerbated suppression of locomotor activity by light (*Figure 11A,B*). In the intact mouse, RAI1-dependend signaling thus seems to limit the light-induced activation of the vSPVZ thereby allowing animals to display activity during the light phase.

Much in the above scenario remains, however, to be demonstrated and species differences rather than the nocturnal versus diurnal niche differentiation might underlie the discrepancy in light-sensitivity between humans and mice lacking one *Rai1* allele. Moreover, a number of other signaling pathways are likely to be involved. Specifically, pathways regulating active waking behaviors might be good alternate candidates. TGFα (transforming growth factor α) activates vSPVZ neurons thereby suppressing locomotor activity without affecting overall time-spent-awake (*Kramer et al., 2001*; *Tournier et al., 2007*), reminiscent of the profound and sustained effects of light in $Rai1^{+/-}$ mice. Likewise, CLC (cardiotrophin-like cytokine) suppresses locomotor activity via CLC receptors on neurons localized in the periventricular region of the hypothalamus (*Kraves and Weitz, 2006*). Both TGF-α and CLC are produced in the SCN in antiphase with the daily (circadian) changes in locomotor activity. Moreover, CLC production is stimulated by light. As was the case for $Rai1^{+/-}$ mice, suppression of locomotor activity by CLC is associated with normal waking behaviors although time-spent-awake and -asleep were not quantified (*Kraves and Weitz, 2006*). Finally, other structures than the vSPVZ have been shown to relay the direct effects of light on behavior such as the intergeniculate leaflet and olivary pretectal nuclei (*Gall et al., 2016*).

## Conclusions

RAI1 is thought to be key to normal brain development and function due to its association with a number of neurobehavioral disorders which, besides SMS, include Potocki–Lupski syndrome (*Potocki et al., 2007*), schizophrenia (*Joober et al., 1999*), autism (*van der Zwaag et al., 2009*), Parkinson disease (*Do et al., 2011*), and cerebellar ataxia (*Hayes et al., 2000*). RAI1 is a highly conserved transcriptional regulator (*Carmona-Mora et al., 2010*) with high neuronal expression (*Toulouse et al., 2003*; *Fragoso et al., 2015*), and recent evidence shows that it acts as a positive transcriptional regulator by binding to active promotor and enhancer regions (*Huang et al., 2016*). Among RAI1 target genes are *Bdnf* (*brain-derived neurotrophic factor*) and *Htr2c* (*serotonin receptor 2c*) (*Huang et al., 2016*; *Burns et al., 2010*), both of which are implicated in hyperphagia leading to obesity, a phenotype accompanying SMS, concomitant with altered locomotor activity (*Huang et al., 2016*; *Burns et al., 2010*; *Kernie et al., 2000*; *Nonogaki et al., 2003*). Moreover, as its name indicates, *Rai1* expression is induced by retinoic acid (*Imai et al., 1995*). Because retinoic acid signaling is a major pathway involved in eye formation (*Cvekl and Wang, 2009*) and other developmental processes, and because RAI1 appears to help assemble and maintain neuronal circuitry (*Huang et al., 2016*), the lack of one *Rai1* allele is likely to impact eye development and light signaling pathways, maybe through its target gene *Bdnf*, which is known to be important for retinal development and synaptic connectivity (*Grishanin et al., 2008*). While we found no apparent structural differences in the retina, the ERG results did reveal a profound attenuation of the response to light pointing to altered retinal function. A more detailed examination of, for example, the morphology of synaptic terminals and expression levels of other photo-transduction proteins could therefore still yield new insights.

The SMS mouse model revealed that both circadian and direct light reception are affected. Our study thus brought new insights into the light-signaling pathways regulating behavior and suggests that *Rai1* is a key regulator in non-visual light perception. Whether these effects are developmental, acute or both, remains to be determined. Given our results, investigating the functional integrity of cells and pathways related to photoreception and light signaling in SMS patients is an obvious next step.

## Materials and methods

### Animals and housing conditions

C57BL/6*Tyrc−Brd* mice heterozygous for the *Rai1* null mutation (*Rai1*$^{+/-}$) were generously provided by James R. Lupski (Baylor College of Medicine, Houston TX, USA). The mutation concerns a 3910 bp deletion due to the insertion of a *neoR*-expression cassette and an *Escherichia coli lacZ* coding sequence into *Rai1* exon 2 (*Bi et al., 2005*). This mutation prevents the encoded protein from entering the nucleus thus resulting in a functionally inactive allele (*Bi et al., 2005*). In the vast majority of cases, homozygosity for this mutation results in embryonic lethality. In all experiments, 10- to 12-week-old male *Rai1*$^{+/-}$ mice along with their male wild-type littermate controls (*Rai1*$^{+/+}$) were used. *Rai1*$^{+/-}$ mice develop obesity by week 20 (*Burns et al., 2010*), but in the current experiments, they were not heavier than their wild-type littermates (*Rai1*$^{+/-}$: 29 ± 0.7; *Rai1*$^{+/+}$: 31 ± 0.3 g). All animals were kept under a 12 hr light/12 hr dark cycle (LD12:12, fluorescent lights, intensity 6.6 cds/m$^2$ or 0.97 μW/cm$^2$) and were singly housed for EEG/EMG and locomotor activity recordings with food and water available *ad libitum*. Under LD12:12 we use Zeitgeber time (ZT) with ZT0 and ZT12 marking light and dark onset, respectively, while under constant dark (DD) conditions circadian time 12 (CT12) was defined as activity onset. Room temperature was kept at 25°C and humidity between 50% and 60%. All experiments were approved by the Ethical Committee of the State of Vaud Veterinary Office, Switzerland.

### Assessment of circadian activity patterns

Animals were housed individually and placed in recording cabinets equipped with passive infra-red (PIR) sensors (Visonic Ltd, Tel Aviv, Israel). ClockLab (ActiMetrics, IL, USA) was used to record and analyze locomotor activity patterns and to schedule the light-dark regimens. Data were sampled at 5-min resolution, and period length (*X2*-periodogram) and onset of the main daily activity phase were extracted to calculate the duration of the active (alpha or α) and rest (rho or ρ) phase under DD conditions. *Rai1*$^{+/+}$ (n = 11) and *Rai1*$^{+/-}$ mice (n = 12; recorded in five separate experimental cohorts from a total of eight different litters) were studied for >10 days under entrained conditions (LD12:12), followed by >20 days of DD 'free-running' conditions, followed again by LD12:12 (>10 days). The expected activity onset of the last day under LD was extrapolated back by linear regression of the DD activity onset times. The difference between the average activity onset under the preceding LD12:12 and this expected activity onset determined the phase angle of entrainment. As a measure of precision of activity onset under DD conditions, we compared the individual correlation coefficients (*r*) of the linear regression of activity onsets explained above to estimate the phase angle of entrainment between genotypes. The same number of observations contributed to each *r*-value, that is 16 circadian days. Correlation coefficients were Fisher's Z-transformed prior to statistical analysis. To estimate robustness of the circadian rhythm under DD, the individually obtained maximal amplitude of the $X^2$-statistic was compared between genotypes as this statistic indexes how much of the total variance in the locomotor activity signal can be attributed to the circadian rhythm. For this analysis, the number of cycles contributing to the analysis was identical for each individual (n = 10/ genotype), that is 18.

Evaluation of the phase-shifting capacity of light was performed in a separate group of mice (*Rai1*$^{+/+}$: n = 5; *Rai1*$^{+/-}$: n = 4; one experimental cohort from a total of two litters). For this experiment, mice were exposed to a 1-hr-light pulse (6.6 cds/m$^2$) at circadian times (CT)16 and −22 (with CT12 designating activity onset). Both light pulses occurred 4 days after the mice were released in DD. The phase shifts were calculated as the difference, in circadian hours, between the average activity onset over the 3 days prior to the light pulse and the average activity onset over the 10 days

following the light pulse. Activity onset on the first day after the light pulse was not taken into account.

In a third group of mice ($Rai1^{+/+}$: n = 5; $Rai1+/-$: n = 3; two experimental cohorts from a total of 3 litters), we tested whether mice are equally sensitive to light of low illumination, that is, 0.6 cds/m$^2$ versus the habitual 6.6 cds/m$^2$, both under a LD12:12 protocol. Mice were exposed to at least 5 days under 6.6 cds/m$^2$ followed by at least 6 days under 0.6 cds/m$^2$. The first day under the new light environment was discarded from the analysis.

## Electroencephalogram (EEG) and electromyogram (EMG) recordings and analysis

EEG/EMG surgeries were performed under deep anesthesia as previously described (*Mang and Franken, 2012*). $Rai1^{+/+}$ (n = 8) and $Rai1^{+/-}$ mice (n = 7; six experimental cohorts from a total of eight litters) were allowed to recover for >10 days before experiments started. EEG and EMG signals were amplified, filtered, analog-to-digital converted, and stored using EMBLA (A10 recorder) and Somnologica software (Medcare Flaga, Thornton, CO). The sleep-wake states wakefulness (W), rapid-eye-movement (REM) sleep, and non-REM (NREM) sleep were visually assigned to consecutive 4 s epochs as previously described (*Mang and Franken, 2012*). Four-second epochs scored as wakefulness in which the EEG was dominated by theta activity (EEG power in the 6.5–9.5 Hz range) were labeled as theta dominated wakefulness (TDW) using an algorithm with the following criteria: if the frequency bin showing highest power density (theta peak frequency or TPF) across the 3.5–15.0 Hz range was within the 6.5–12.0 Hz range, and the ratio of theta power in the ±1 Hz range surrounding the TPF over the total power across 3.5–45.0 Hz was above 0.228, then that epoch would score as TDW, if an additional set of three contextual criteria for that epoch were met: (1) the epoch immediately preceding a TDW epoch was scored as 'W' or 'TDW'; (2) the epoch immediately following a TDW epoch was not scored as NREM sleep; (3) isolated TDW epochs preceded and followed by 3 'W' epochs were excluded. Recordings of three C57BL/6J male mice were assessed visually for TDW state scoring and the 0.228 theta/total power ratio threshold was chosen because it yielded the best match with visually identified TDW. Please see (*Vassalli and Franken, 2017*) for details and validation. The term TDW was previously coined to describe a theta-rich waking sub-state associated with locomotor activity in the mouse (*Welsh et al., 1985*).

EEG/EMG signals were recorded continuously for a 3-day period, the first two of which served as baseline (48 hr) under LD12:12. During the first 6 hr of day 3 (ZT0-6), animals were sleep deprived by 'gentle handling' (*Mang and Franken, 2012*). Locomotor activity was recorded in parallel using the same PIR sensors. In a separate cohort (n = 4/genotype from a total of 2 litters), activity and sleep-wake patterns were investigated under short light-dark cycles (LD1:1; 1 day under LD12:12 followed by 2 days of LD1:1) and constant light (LL; 1 day under LD12:12 followed by 2 days under LL) conditions. The two experiments were separated by 3 days under LD12:12. In two individuals per genotype, behavior was further verified using infra-red video recordings.

## Electroretinogram (ERG) recording and analysis

For scotopic conditions, mice ($Rai1^{+/+}$: n = 5; $Rai1^{+/-}$: n = 6; 15 weeks old; two experimental cohorts from a total of four litters) were dark-adapted overnight and experiments were done under dim-red light. Corneal ERG was recorded with an active ring electrode and responses to single flashes of green light (520 nm; half-bandwidth 35 nm; at 10–4, 10–3, 10–2, 3 × 10–2, 10–1, 3 × 10–1, 1, 3, 10, and 30 cds/m$^2$), generated by a stroboscope in the upper part of a Ganzfeld stimulator (Espion E3 apparatus; Diagnosys LLC, Lowell, MA), were quantified. For each stimulus intensity, 10 to 15 traces were averaged for each eye separately. The $a$-wave amplitude (photoreceptor response) was defined as the difference between the peak of the $a$-wave and baseline at the time of stimulation. The $b$-wave amplitude (second-order neurons) was defined as the difference between the peak of the $b$-wave and the peak of the $a$-wave.

## Retinal histology

Eye collection, retina processing, and immunolabeling were performed as previously described (*Bemelmans et al., 2006*). Specifically, antibodies directed against Rhodopsin (1:500, #MA5-11741, Pierce), G-Protein Subunit Alpha Transducin 2 (GNAT2, 1:200, RRID:AB_2279097, #Sc-390, Santa

Cruz, Santa Cruz, CA), Protein kinase C alpha (PKCα, 1:200, RRID:AB_2168560, #sc-10800, Santa Cruz), Glial fibrillary acidic protein (GFAP, 1:500, RRID:AB_10013382, #Z0334, Dako, Zug, Switzerland), melanopsin (1:500, RRID:AB_444884, #Ab19383, Abcam, Cambridge, UK), calbindin (1:5000, RRID:AB_10000340, CB-38a, Swant, Marly, Switzerland), ChAT (1:2500, polyclonal goat, gift from Jean-Pierre Hornung, University of Lausanne), and GABA (1:500, RRID:AB_477652, #2052, Sigma, Buchs, Switzerland) were used and visualized with the appropriate secondary antibodies coupled to Alexa Fluor488 or Alexa633 (1:2000, Molecular Probes, Eugene, OR).

## c-FOS immunofluorescence

Mice were either exposed to a 1-hr-light pulse from ZT14.5 to ZT15.5 (6.6 cds/m$^2$; n = 6 and 7 for wild-type and heterozygous mice, respectively) or left undisturbed (controls; n = 3/genotype) and sacrificed immediately at ZT15.5. Four experimental cohorts were used comprising nine litters. Mice were deeply anesthetized with pentobarbital sodium (300 mg/ml diluted at 1:15) administered intraperitoneally in a volume of 10 ml/kg body weight. Animals were then perfused transcardially with heparin/PBS followed by a 15 min fixation with 4% paraformaldehyde in PBS. In control mice, this was performed under dim-red light. Brains were carefully removed and post-fixed overnight in the fixative at 4°C. Cryoprotection of brains was achieved by using graded concentrations of sucrose (10-, 20-, and 30%). After 2 days of dehydration at 4°C in sucrose, brains were rapidly frozen in isopentane (Sigma-Aldrich Co, MO) vials placed on dry ice and stored at −80°C. Coronal sections (25 μm) were cut at −20°C using a cryostat. SCN sections were collected (0.22–0.83 mm posterior to bregma). Cryostat sections were mounted and left to air dry for at least 1 hr before being washed in PBS/0.25% Triton and incubated overnight with an anti-c-FOS antiserum (1:250; rabbit polyclonal, Santa Cruz Biotechnology Inc, CA). An Alexa568-conjugated goat anti-rabbit antibody was used as secondary antibody (1:1000, Molecular Probes, Thermo Fisher Scientific Inc, MA) to visualize c-FOS staining. DAPI staining (1:10000, Sigma-Aldrich Co, MO) was used to visualize cell nuclei. Sections were examined and photographed with a Zeiss Axio imager M1 microscope. The number of c-FOS-positive neurons as well as the intensity of the immunoreactivity was quantified using the ImageJ/FIJI plugin (National Institute of Health, MD) for the left and right hemispheres separately.

## Gene expression in retina and brain

Pooled mRNA from left and right retina was isolated (n = 5/genotype; one experimental cohort, a total of 3 litters) and purified using the TRIzol RNA Isolation Reagents protocol (Invitrogen, Carlsbad, CA). RNA quantity was assessed with a NanoDrop ND-1000 spectrophotometer (ThermoScientific, Wilmington, DE). For cDNA synthesis, 0.5 μg of total RNA was treated with DNaseI (Amplification Grade, Invitrogen, Carlsbad, CA), reverse-transcribed using random hexamers and Superscript II reverse transcriptase (Invitrogen, Basel, Switzerland) according to standard procedures. The cDNA was diluted 10 times and 1 μl was used as template in a 10 μl TaqMan reaction on an ABI PRISM HT 7900 detection system in technical triplicates. Forward and reverse primers, as well as probes sequences were chosen from the coding regions of genes in Invitrogen website (*Rai1, Opn1mw, Opn4, Rho,* and *Thy1*). The expression of each gene was normalized against four control reference transcripts (*ActinB, Eef1a1, Rps9,* and *TubA*) using Qbase (*Hellemans et al., 2007*).

To assess the 24 hr time course of *Rai1* brain expression under LD12:12 conditions, samples were collected at 3 hr intervals. At each interval, three male C57BL/6J mice were sacrificed by decapitation after being anesthetized with isoflurane, and the forebrain (whole brain minus brainstem and cerebellum) was dissected. Sampling was performed under normal light conditions during the light phase (ZT0, −6, and −9), and under dim red light during the dark (ZT12, −15, and −18). For each time point, a separate experimental cohort was used originating from four different litters. The tissue was placed in an Eppendorf tube, immediately flash frozen in liquid nitrogen and kept at −80°C until RNA extraction. RNA was isolated and purified with the RNeasy Lipid Tissue Midi kit (Qiagen, Hombrechtikon, Switzerland) and treated with DNaseI. Assessment of RNA quantity, cDNA synthesis, and amplification as described above. As reference transcripts *Eef1a1, Gapdh, Gusb,* and *Rps9* were used. Time courses of *Arntl* (*Bmal1*) and *Per2* expression were used as positive control.

## Statistics

Statistical analyses were performed using R Statistical Software (Foundation for Statistical Computing, Vienna, Austria) and SAS (SAS Institute Inc, Cary, NC). Significant effects of genotype and time-of-day and their interactions were assessed using analysis-of-variance (ANOVA) and decomposed using post-hoc $t$-tests. Simple two-group genotype contrasts were assessed using $t$-tests. Circadian rhythmicity in locomotor activity was evaluated using chi-square periodogram analysis (ClockLab, Actimetrics). Statistical significance was set to $p < 0.05$ and results are reported as mean $\pm$ standard error of the mean (SEM). SigmaPlot 11 (Systat Software Inc., Chicago, IL) was used to generate graphs.

## Acknowledgements

We thank Mirjam Münch for her constructive and important input on the project, Alexandre Reymond for housing the mice and sharing his expertise on genomic structural changes and their resulting pathologies, James R Lupski for generously providing the $Rai1^{+/-}$ mice, Yann Emmenegger, Sylvain Crippa, and Catherine Moret for excellent technical support, Jean-Pierre Hornung for providing the ChaT antibodies, H Craig Heller for helpful discussions, and Charlotte Hor for editing the manuscript.

## Additional information

### Funding

| Funder | Grant reference number | Author |
| --- | --- | --- |
| Schweizerischer Nationalfonds zur Förderung der Wissenschaftlichen Forschung | CRSII3_136201 | Paul Franken |
| State of Vaud, Switzerland | | Paul Franken |
| Schweizerischer Nationalfonds zur Förderung der Wissenschaftlichen Forschung | 31003A_146694 | Paul Franken |

The funders had no role in study design, data collection and interpretation, or the decision to submit the work for publication.

### Author contributions

SD, Conceptualization, Formal analysis, Supervision, Investigation, Visualization, Writing—original draft, Writing—review and editing; CK, Formal analysis, Investigation, Visualization, Writing—review and editing; YA, Conceptualization, Supervision, Writing—review and editing; AK, Conceptualization, Writing—review and editing; PF, Conceptualization, Formal analysis, Supervision, Funding acquisition, Visualization, Writing—original draft, Project administration, Writing—review and editing

### Author ORCIDs

Paul Franken, http://orcid.org/0000-0002-2500-2921

### Ethics

Animal experimentation: All experiments were approved by the Ethical Committee of the State of Vaud Veterinary Office, Switzerland (# VD2545).

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
