## [Decision Letter]

Thank you for submitting your article "*Rai1* Frees Mice from the Repression of Active Wake Behaviors by Light" for consideration by *eLife*. Your article has been reviewed by three peer reviewers, one of whom, Louis J Ptáček (Reviewer #1), is a member of our Board of Reviewing Editors, and the evaluation has been overseen by a Senior Editor. The following individuals involved in review of your submission have agreed to reveal their identity: Ying-Hui Fu (Reviewer #2) and Mark S Blumberg (Reviewer #3).

The reviewers have discussed the reviews with one another and the Reviewing Editor has drafted this decision to help you prepare a revised submission.

Summary:

SMS is a syndrome in which patients may manifest obesity, aggressive behavior and craniofacial and skeletal dysmorphisms. In addition, they have sleep phase advance and frequent nocturnal awakening and excessive daytime sleepiness. The manuscript by Diessler and colleagues describes study of a mouse model of SMS to better understand the sleep phenotype. The SMS mutations cause haploinsufficiency for *Rai1*. This paper provides a thorough and largely compelling assessment of the effects of *Rai1* haploinsufficiency on circadian and sleep biology, light sensitivity, and other related processes in adult mice. The authors should be commended on their thoroughness, although at times I felt that the core narrative was lost in the pursuit of too much text devoted to subsidiary issues. Where the paper is strongest and most convincing is in the assessments of sleep-wake behavior and electrophysiology, as well as the hyperresponsivity of the mice to light. Because humans with *Rai1* mutations exhibit circadian and light responses that are opposite to those of the mice (e.g., light increases activity in humans rather than suppressing it, as in the mice), the authors are led to consider the very interesting idea that species differences on the diurnal-nocturnal dimension accounts for the opposite effects. Building on recent work in nocturnal Norway rats and diurnal Nile grass rats, they provide data and a model to explain the human-mouse differences as related to how *Rai1* haploinsufficiency affects activity in the vSPVZ. This last finding highlights the need to attend to species differences when using mice to model human conditions.

Essential revisions:

1) The authors report *Rai1* gene expression in retina and brain by sampling every 3 hours. How does this data compare with that in the public domain looking at global expression analysis in different tissues? Is *Rai1* expressed in SCN and/or vSPVZ, or surrounding areas?

2) Given the important observations re: changes in sensitivity to entrainment, it is surprising that the authors did not present full PRCs. This would be worthwhile if it could be done in a timely fashion.

3) The authors discuss the fact that the mouse model of SMS does not fully recapitulate the human phenotype, vis-à-vis behavioral outputs. This might, as they mention, represent differences between nocturnal and diurnal animals. Alternately, it may reflect differences in human vs. mouse physiology. The same conundrum has been noted in some of the mouse models of human FASP mutations.

4) The model which it seems the authors favor should be explicitly articulated. Under normal condition, *Rai1* protein seems to serve as inhibitor for vSPVZ (-0.7mm) activation by light. vSPVZ activation leads to "suppression of activity in light" based on results from previous reports. *Rai1*+/+ can inhibit this vSPVZ's suppression of behavior, so animals have some activities in light when in need. *Rai1* +/- has reduced inhibition of vSPVZ. This means these mice have higher vSPVZ activities, which further enhances suppression of movement in light, therefore have almost no activity in light.

5) There are a number of problems with the Fos data they present in Figure 10. First, they claim to see a difference in Fos-positive cells between genotypes in the vSPVZ only at the section which is -.71 mm from bregma, yet the only representative Fos data they present (in Figure 10) is of the -.83 mm section (as stated in the figure legend). Second, they do not provide the scale bars in 10A or 10B, so it is hard to determine where the vSPVZ "box" as depicted in 10A would be in 10B. Third, it is very hard to see Fos in the area above the SCN in the images in 10B -- a higher magnification inset would help. Finally, many people who've studied the vSPVZ would say you need to show either VIP or AVP immuno to demonstrate you are actually outside the SCN and that you know how to differentiate the two structures (the vSPVZ is traditionally defined as either the area innervated by VIP fibers from the SCN or as the area dorsal to AVP neurons of the SCN "shell" and ventral to the AVP neurons of the PVN, or both).

6) Another issue with the Fos experiment concerns the sufficiency of the sample size: N = 3/genotype, as stated in the text. Given the inherent variability of Fos immunoreactivity, this is too small a sample size to feel confident that the results would replicate. And given that this finding is the basis for the model in Figure 11, it seems reasonable to request that additional subjects be tested (and the findings presented more comprehensively; see previous paragraph) to ensure that this aspect of the study is reliable. The sample size in the Fos study is among the smallest of all the experiments. In general, Ns range from 11-12 in the behavioral and electrophysiological experiments to Ns of 3-6 in some of the subsidiary experiments. The problem of small Ns is illustrated very clearly in [Supplementary-material SD2-data], in which a finding with P = 0.04 is considered significant whereas two other findings with P = 0.06 is described like this: "for none of the three sleep-wake states (i.e., waking, NREM sleep, and REM sleep) did the 24h values significantly differ ([Supplementary-material SD2-data])." This is technically correct, but the sample size for these comparisons was only 4/genotype. Such small-N comparisons with borderline p values should be viewed with skepticism.

---

## [Author Response]

*Essential revisions:*

*1) The authors report Rai1 gene expression in retina and brain by sampling every 3 hours. How does this data compare with that in the public domain looking at global expression analysis in different tissues? Is Rai1 expressed in SCN and/or vSPVZ, or surrounding areas?*

*Rai1* is expressed throughout the brain (Toulouse et al., Genomics 2003; Bi et al., 2007; Fragoso et al., 2015; Huang et al., Neuron 2016) and we now have specified in the Results section that this expression is wide-spread and especially pronounced in olfactory bulb, cerebral cortex, hippocampus, striatum, thalamus, hypothalamus, cerebellum, and brainstem. Public databases such as the Expression Atlas (www.ebi.ac.uk/gxa/genes/), the Allan Brain Atlas (http://mouse.brain-map.org), and MGI’s gene expression database (GXD; http://www.informatics.jax.org/expression.shtml) confirm this. *Rai1* expression is also present in several peripheral tissues (liver, lung, digestive system), especially during development.

Although *Rai1* is expressed in the hypothalamus we are not aware of publications in which its expression dynamics was quantified specifically in the SCN. The *CircaDB* (http://circadb.hogeneschlab.org/) database includes, however, a number of time-course transcriptome data sets, 5 of which concern the SCN. In 1 out of these 5, *Rai1* expression is deemed rhythmic but this result depend on the type of periodogram analyses chosen and does not survive a 5% false-discovery rate. In none of the other brain areas included in this database does *Rai1* expression vary over time thereby confirming the observations made in our study. In the Results section we now refer to the *CircaDB* database. Studies specifically focusing on the vSPVZ are not available to our knowledge.

*2) Given the important observations re: changes in sensitivity to entrainment, it is surprising that the authors did not present full PRCs. This would be worthwhile if it could be done in a timely fashion.*

This was one of two additional experiments suggested by the reviewers. Even with the granted 1 month extension, 3 months is still tight to accomplish these types of experiments especially considering the low breeding yield of *Rai1+/-* mice. Because of this low breeding yield the data in the manuscript now took close to 4 years to collect. We estimate that performing both experiments would take close to a year, as we currently do not have sufficient mice of the right age, sex, and genotype, despite having considerably expanded the breeding of this line. Because a full PRC, to us, seems less pertinent to support the main claims made in the paper, requires more mice, and because the reviewers stated that it “would be worthwhile if it could be done in a timely fashion”, we used all suitable mice (7) to increase sample size for the second experiment instead (see point 6).

*3) The authors discuss the fact that the mouse model of SMS does not fully recapitulate the human phenotype, vis-à-vis behavioral outputs. This might, as they mention, represent differences between nocturnal and diurnal animals. Alternately, it may reflect differences in human vs. mouse physiology. The same conundrum has been noted in some of the mouse models of human FASP mutations.*

Indeed. We now have specifically pointed out this possibility in the Discussion (see “Reconciling species differences…” paragraph).

*4) The model which it seems the authors favor should be explicitly articulated. Under normal condition, Rai1 protein seems to serve as inhibitor for vSPVZ (-0.7mm) activation by light. vSPVZ activation leads to "suppression of activity in light" based on results from previous reports. Rai1+/+ can inhibit this vSPVZ's suppression of behavior, so animals have some activities in light when in need. Rai1 +/- has reduced inhibition of vSPVZ. This means these mice have higher vSPVZ activities, which further enhances suppression of movement in light, therefore have almost no activity in light.*

We have put in an effort to make the Discussion section concerned (see “Reconciling species differences…” paragraph) more straightforward and have more clearly formulated our ideas. Apart from an acute and active signaling role for *RAI1* as a transcriptional regulator, other scenarios could be envisioned such as one involving *RAI1* in establishing and maintaining the appropriate signaling circuitry. Such role has been demonstrated for *RAI1*’s role in another hypothalamic network involved in food intake (Huang et al., 2016). Our present data do not allow to favor either possibility (acute signaling *versus* establishing the correct neuronal circuitry) as stated in the Conclusion section.

*5) There are a number of problems with the Fos data they present in Figure 10. First, they claim to see a difference in Fos-positive cells between genotypes in the vSPVZ only at the section which is -.71 mm from bregma, yet the only representative Fos data they present (in Figure 10) is of the -.83 mm section (as stated in the figure legend). Second, they do not provide the scale bars in 10A or 10B, so it is hard to determine where the vSPVZ "box" as depicted in 10A would be in 10B. Third, it is very hard to see Fos in the area above the SCN in the images in 10B -- a higher magnification inset would help. Finally, many people who've studied the vSPVZ would say you need to show either VIP or AVP immuno to demonstrate you are actually outside the SCN and that you know how to differentiate the two structures (the vSPVZ is traditionally defined as either the area innervated by VIP fibers from the SCN or as the area dorsal to AVP neurons of the SCN "shell" and ventral to the AVP neurons of the PVN, or both).*

We redid this experiment (see point 6) and the figures. We have highlighted the scale bars and put in an effort to make the figures clearer. The genotype effect on c-FOS in the vSPVZ, although significant, is not immediately apparent by eye due to the relative sparsity of c-FOS positive cells (as compared to the SCN) and the somewhat larger variability.

We are, however, confident not to confuse c-FOS reactivity (red dots in all 4 panels of Figure 12) within the SCN with that quantified in the vSPVZ. First the compactness of the DAPI-stained SCN cell nuclei (left two panels) reliably delimits the extent of the SCN which we now confirmed by adding AVP staining in the new experiment (labelled green; right two panels; upper panels matching images of a *Rai1+/-* brain at bregma -0.83mm; lower panels of a *Rai1+/+* brain at -0.73mm). Although the quality of AVP staining could not be optimized, its expressing in the shell follows the DAPI-derived outline of the SCN. Moreover, it can be clearly seen that the c-FOS expression in the SCN is largely limited to its retinorecipient core located ventrally (devoid of AVP staining) precluding mixing-up c-FOS induced in the SCN with that of the sSPVZ. See also Schwartz et al., Neuroscience2000 and Abrahamson & Moore Brain Res2001, and our comparable analysis performed in *Opn4-/-* mice (Tsai et al., PLoS Biol2009).

Author response image 1.**DOI:**
http://dx.doi.org/10.7554/eLife.23292.023

The new data confirmed our original claim that *Rai1+/-* mice have less cFOS reactivity in the SCN and more in the vSPVZ area upon light exposure. However, the claim concerning section-specificity of the genotype effect within the vSPVZ (the difference between the two sections referred to as -0.71mm and -0.83mm) does no longer hold; within the new subset values were significantly higher in the -0.83mm section and not, as we previously reported, limited to the -0.71mm section. Our techniques are apparently not sensitive enough to reliably distinguish between the two. In the new analyses we therefore pooled the number of cFOS+ cells in both sections within each animal and removed the claim of this specificity in the manuscript. For consistency we also pooled the results of the SCN which showed robust and significant decreases in both sections and in both experiments. As the two sections on which this analysis was based differ in SCN size we still show examples of both sections in Figure 10.

*6) Another issue with the Fos experiment concerns the sufficiency of the sample size: N = 3/genotype, as stated in the text. Given the inherent variability of Fos immunoreactivity, this is too small a sample size to feel confident that the results would replicate. And given that this finding is the basis for the model in Figure 11, it seems reasonable to request that additional subjects be tested (and the findings presented more comprehensively; see previous paragraph) to ensure that this aspect of the study is reliable. The sample size in the Fos study is among the smallest of all the experiments. In general, Ns range from 11-12 in the behavioral and electrophysiological experiments to Ns of 3-6 in some of the subsidiary experiments. The problem of small Ns is illustrated very clearly in [Supplementary-material SD2-data], in which a finding with P = 0.04 is considered significant whereas two other findings with P = 0.06 is described like this: "for none of the three sleep-wake states (i.e., waking, NREM sleep, and REM sleep) did the 24h values significantly differ ([Supplementary-material SD2-data])." This is technically correct, but the sample size for these comparisons was only 4/genotype. Such small-N comparisons with borderline p values should be viewed with skepticism.*

Indeed an n of 3 is small and the request for additional mice is very reasonable. As explained to the Editor upon receiving the reviews (see also point 2) we could include 7 additional mice (3 wt, 4 hets) that we successfully used for the c-FOS experiment, thereby raising the total n’s to 6 and 7, in *Rai1+/+* and *Rai1+/-* mice, respectively. These numbers are within range of what is current in these types of experiments (e.g., Todd, Gall, Weiner and Blumberg, 2012). By confirming the overall effects we first described (but not the section specificity; see main point 5), we now feel confident about the results and thank the reviewers for suggesting the additional experiment.

More generally, the number of mice used is always a compromise among many considerations such as having enough statistical power while being urged to reduce the number of animals by animal-use committees. The initial, larger experiments demonstrated that the genotype differences in the effects of light on locomotor activity and theta-dominated waking were large. We therefore thought it justifiable that for follow-up experiments, in which we were able to reproduce the initial observations, to use fewer mice. The main behavioral findings on which our manuscript is based are very robust and can be readily observed in each individual.